# An Online Feasible Point Method for Benign Generalized Nash Equilibrium Problems

**Sarah Sachs**                                                SARAH.C.SACHS@GMAIL.COM
*Department of Computing Sciences, Bocconi University*

**Hédi Hadiji**                                                HEDI.HADIJI@GMAIL.COM
*Laboratoire des signaux et systèmes, Univ. Paris-Saclay, CNRS, CentraleSupélec*

**Tim van Erven**                                                TIM@TIMVANERVEN.NL
*Korteweg-de Vries Institute for Mathematics, University of Amsterdam*

**Mathias Staudigl**                          MATHIAS.STAUDIGL@MANNHEIMUNIVERSITY.DE
*School of Business Informatics and Mathematics, University of Mannheim*

**Editors:** Gautam Kamath and Po-Ling Loh

## Abstract

We consider a repeatedly played generalized Nash equilibrium game. This induces a multi-agent online learning problem with joint constraints. An important challenge in this setting is that the feasible set for each agent depends on the simultaneous moves of the other agents and, therefore, varies over time. As a consequence, the agents face time-varying constraints, which are not adversarial but rather endogenous to the system. Prior work in this setting focused on convergence to a feasible solution in the limit via integrating the constraints in the objective as a penalty function. However, no existing work can guarantee that the constraints are satisfied for all iterations while simultaneously guaranteeing convergence to a generalized Nash equilibrium. This is a problem of fundamental theoretical interest and practical relevance. In this work, we introduce a new online feasible point method. Under the assumption that limited communication between the agents is allowed, this method guarantees feasibility. We identify the class of benign generalized Nash equilibrium problems, for which the convergence of our method to the equilibrium is guaranteed. We set this class of benign generalized Nash equilibrium games in context with existing definitions and illustrate our method with examples.

## 1. Introduction

In this work, we study multi-agent systems with shared constraints. Our goal is to understand the guarantees with respect to the convergence to an equilibrium and the feasibility of the repeated interactions between strategic and independent players. If satisfying the shared constraint is indispensable, it is necessary that the agents cooperate. To enable this cooperation, we consider a setting where limited communication between agents is possible. For each player, this induces an interesting trade-off between strategic myopic optimization of the losses and cooperation to guarantee feasibility. More precisely, we consider the repeated interaction of $n$-players on a *generalized Nash-equilibrium problem* (GNEP) introduced by Arrow and Debreu (1954), see Definition 1. The difference to a standard Nash equilibrium problem is that the players share joint constraints. Hence, in an $n$-player game, the feasibility of the $i^{\text{th}}$-players action $x^{(i)}$ may depend on the choices of all other players $x^{(-i)} = [x^{(1)}, \ldots, x^{(i-1)}, x^{(i+1)}, \ldots x^{(n)}]$. We define a protocol for the repeated interaction of $n$-players which allows for some limited communication between the players. More specifically, before committing to the next iterate $x_{t+1}^{(i)}$, players are allowed to communicate a *desired set* $\mathcal{S}_{t+1}^{(i)}$. These sets indicate a region within which the players want to play. For each iteration $t$, and each player $i$:

1. Selects a desired set $\mathcal{S}_t^{(i)}$, and sends it to all other players. Receives their sets $\mathcal{S}_t^{(j)}$ for $j \in [n] \setminus \{i\}$,

2. Chooses iterate $x_t^{(i)}$ depending on $\mathcal{S}_t^{(i)}$ and $\mathcal{S}_t^{(j)}$'s,

3. Suffers loss with respect to the loss function. If the shared constraints are violated, pay $+\infty$.

In this work, we give an algorithm for the updates of the desired sets $\mathcal{S}_t^{(i)}$ and the iterates $x_t^{(i)}$. Our contributions are as follows:

1. We introduce the *online feasible point method* (online FPM): a new distributed algorithm for computing approximate GNEs. Under the assumption that each player uses the online FPM in a repeated generalized game, we show in Theorem 7 that feasibility is guaranteed for each iteration. This online FPM follows a fundamentally different approach from existing methods.

2. We identify a subclass of GNEP, which we name strongly benign GNEP, for which convergence of the iterates to an equilibrium is guaranteed. This is shown in Theorem 8. Furthermore, we illustrate the class of strongly benign GNEP with examples and set it into context with common assumptions from the literature.

3. In Section 5, Theorem 9, we derive regret bounds for each player and set these results in context with existing results on online learning with varying constraints. We note that for online problems with varying constraints, strong guarantees are possible if the constraints are not adversarial but endogenous to the system.

4. We illustrate the behavior of the online FPM with various examples of strongly benign GNEP. Furthermore, we demonstrate with experiments that our algorithm convergence to the GNEP without constraint violation for GNEP beyond strongly benign GNEP.

**Practical Relevance.** A practically relevant example of a GNEP where feasibility is desirable is the repeated interaction of agents (sellers and buyers) in the electricity market. Most power grids, e.g., the European power grid, impose strong restrictions on the fluctuation of the Hertz frequency to avoid risking a blackout. Since a nationwide blackout comes at a very high cost for all agents, providing feasibility guarantees is indispensable (see Saad et al. (2012) for more details). Further application examples where feasibility guarantees are desirable can be found in pollution control or economic models (see Facchinei and Kanzow (2007) for more details).

**Related Work.** The generalized Nash-equilibrium problem was first introduced by Arrow and Debreu (1954). Learning of generalized Nash Equilibria (GNE) so far relied on a primal-dual approach in which Lagrangian-based splitting schemes are constructed. This was pioneered by Yi and Pavel (2019), and further studied in a distributed setting (Liakopoulos et al., 2019; Cao and Liu, 2019; Jordan et al., 2023; Franci et al., 2020). Penalty methods are a further toolbox that enforces the feasibility of the limit solution (Facchinei and Kanzow, 2010; Kanzow et al., 2019; Sun and Hu, 2021). However, these methods do not necessarily guarantee the feasibility of all iterates in a distributed setting and thus differ in techniques and theoretical guarantees. Our work is also related to the flourishing literature on online learning in games (Rakhlin and Sridharan, 2013; Daskalakis et al., 2021, 2011). For zero-sum games without shared constraints, Rakhlin and Sridharan (2013) provide convergence guarantees for optimistic online methods. In the context of online learning,

there is a growing interest in regret guarantees for online learning problems with non-stationary constraints (Neely and Yu, 2017; Liu et al., 2021; Kolev et al., 2023). In contrast to this work, we consider moving constraints that are endogenous to the system rather than adversarially chosen. This makes our setting strictly easier and hence permits us to provide stronger guarantees. The analysis of the online FPM builds on an inexact gradient descent analysis and is, therefore, related to results on inexact gradient methods from the mathematical optimization literature Nesterov (2015); Bertsekas and Tsitsiklis (2000); Khanh et al. (2024).

**Outline.** In Section 2, we introduce the main objective of our work: the generalized Nash equilibrium problem. We furthermore introduce the subclass of strongly benign and benign generalized Nash-equilibrium problems and set this subclass in context to existing subclasses of generalized games. In Section 3, we introduce our algorithm: an online feasible point method. We illustrate the convergence behavior of this algorithm via experiments on several benign and non-benign generalized games. In Section 4, we analyze the convergence of the online feasible point method. For benign generalized Nash equilibrium problems, we show convergence of the iterates to the equilibrium while preserving feasibility for all iterations. In Section 5 we relate our findings to regret analysis: we derive regret bounds for the individual players with respect to the loss and the constraint violations. Hence providing robustness guarantees with respect to the incentive of an individual player to deviate from its strategy. We also discuss the limitations of these guarantees together with further research directions in Section 6.

**Notation.** Throughout the paper, we denote indices for iterates by subscripts, e.g., $x_t$ or $\mathcal{S}_t$, and indices for players by superscripts in brackets, e.g., $x^{(i)}$ or $\mathcal{S}^{(i)}$. Following standard notation we denote $[a, b] = \{a, a+1, \ldots, b\}$ and $[b] = \{1, \ldots, b\}$ for $a, b \in \mathbb{N}$ with $a \leqslant b$. Thus, for an $n$-player game $[n] = \{1, \ldots, n\}$ denotes the set of indices for all players. For sets $\mathcal{A}, \mathcal{B}$, we let $\mathcal{A} + \mathcal{B}$ denote the Minkowski addition, that is, $\mathcal{A} + \mathcal{B} = \{a + b : a \in \mathcal{A}, b \in \mathcal{B}\}$. Following common notational convention, we denote the players' choices of all except the $i^{\text{th}}$ player by $x^{(-i)}$ and similarly the product over all sets except the $i^{\text{th}}$ by $\mathcal{S}^{(-i)} = \prod_{j \in [n] \setminus \{i\}} \mathcal{S}^{(j)}$. Similarly, we denote the product space over all $n$ players by $\mathcal{S}_t^{[\times [n]]} = \prod_{i \in [n]} \mathcal{S}_t^{(i)}$. Throughout the paper, we let $\|\cdot\|$ denote the Euclidean norm. For any compact set $\mathcal{X} \subset \mathbb{R}^n$, we denote the diameter of a set as $\text{Diameter}(\mathcal{X}) = \max_{x, y \in \mathcal{X}} \|x - y\|$ and the distance of any $u \in \mathbb{R}^n$ to $\mathcal{X}$ as $\text{dist}(\mathcal{X}, u) = \min_{x \in \mathcal{X}} \|x - u\|$. Further, for compact sets $\mathcal{X}, \mathcal{A} \subset \mathbb{R}^n$, we let $\text{dist}(\mathcal{X}, \mathcal{A}) = \min_{u \in \mathcal{A}} \text{dist}(\mathcal{X}, u)$. Following standard notation convention, we let $\Delta_k \subseteq \mathbb{R}^k$ denote the $k$-dimensional standard simplex, that is $\Delta_k = \{x \in \mathbb{R}^k : x_i \geqslant 0, \sum_{i=1}^{k} x_i = 1\}$. Further, for any $x \in \mathbb{R}^d$ and $a > 0$, we let $\text{B}_2(x, a) := \{z \in \mathbb{R}^d : \|x - z\|_2 \leqslant a\}$ denote the Euclidean ball with radius $a$ and center $x$.

## 2. Generalized Nash Equilibrium Problem

In this section, we formally introduce our main objective. Namely, the generalized Nash-equilibrium problem and the subclass of benign generalized Nash-equilibrium problems.

### 2.1. Setting

In a repeatedly played generalized Nash-equilibrium problem, each player $i \in [n]$ has control over a variable $x^{(i)} \in \mathbb{R}^{d^{(i)}}$, where $d^{(i)} \in \mathbb{N}$ denotes the dimension of the action space of player $i$. We denote $d^{([n])} = \sum_{i \in [n]} d^{(i)}$, and $d^{(-i)} = d^{([n])} - d^{(i)}$. We let $\mathfrak{C} \in \mathbb{R}^{d^{([n])}}$ denote the shared

constraint set for the players. That is, we require that $[x^{(1)}, \ldots x^{(n)}]^{\top} \in \mathfrak{C}$. Hence, for player $i$, the set of feasible actions, given the choice of actions of the other players, is

$$\mathfrak{X}^{(i)}\left(x^{(-i)}\right) := \left\{x^{(i)} \in \mathbb{R}^{d^{(i)}} \mid \left(x^{(i)}, x^{(-i)}\right) \in \mathfrak{C}\right\}.$$

The objective of each player $i \in [n]$ is to minimize their loss $\nu^{(i)} : \mathbb{R}^{d^{(i)}} \times \mathbb{R}^{d^{(-i)}} \to \mathbb{R}$. With this notation, we introduce the generalized Nash equilibrium problem. The aim of each player $i \in [n]$ in a *Generalized Nash Equilibrium Problem* (GNEP) is to minimize its loss. That is

$$\min_{x^{(i)}} \nu^{(i)}\left(x^{(i)}, x^{(-i)}\right) \quad \text{s.t. } x^{(i)} \in \mathfrak{X}^{(i)}(x^{(-i)}).$$

Throughout the paper, we assume boundedness and convexity of the constraint set and strong convexity of the loss functions:

**Assumption 2.1**

1. $\mathfrak{C}$ *is non-empty, compact and convex.*

2. *For all $i \in [n]$ and $x^{(-i)} \in \mathbb{R}^{d^{(-i)}}$, the function $\nu^{(i)}(\,\cdot\,, x^{(-i)}) : \mathbb{R}^{d^{(i)}} \to \mathbb{R}$ is differentiable on $\mathbb{R}^{d^{(i)}}$ and $\mu$-strongly convex for $\mu > 0$. That is, for any $x^{(-i)} \in \mathbb{R}^{d^{(-i)}}$ it holds for all $x^{(i)}, \tilde{x}^{(i)} \in \mathfrak{X}^{(i)}(x^{(-i)})$ that*

$$\nu^{(i)}\left(x^{(i)}, x^{(-i)}\right) - \nu^{(i)}\left(\tilde{x}^{(i)}, x^{(-i)}\right) \leqslant \left\langle \nabla_{x^{(i)}} \nu^{(i)}\left(x^{(i)}, x^{(-i)}\right), x^{(i)} - \tilde{x}^{(i)} \right\rangle - \frac{\mu}{2} \left\| x^{(i)} - \tilde{x}^{(i)} \right\|^2.$$

3. *For all $i \in [n]$ and $x^{(i)} \in \mathbb{R}^{d^{(i)}}$, the function $\nu^{(i)}(x^{(i)}, \,\cdot\,) : \mathbb{R}^{d^{(-i)}} \to \mathbb{R}$ is continuous.*

4. *For all $i \in [n]$, the gradient $\nabla_{x^{(i)}} \nu^{(i)}\left(x^{(i)}, x^{(-i)}\right)$ is bounded for all $(x^{(i)}, x^{(-i)}) \in \mathfrak{C}$, i.e., $\left\| \nabla_{x^{(i)}} \nu^{(i)}\left(x^{(i)}, x^{(-i)}\right) \right\| \leqslant G$. We further assume that $G$ is known to the players.*

For the benefit of readability, we denote $\nabla_{x^{(i)}} \nu^{(i)}\left(x^{(i)}, x^{(-i)}\right)$ by $\nabla_i \nu^{(i)}\left(x^{(i)}, x^{(-i)}\right)$. As for standard games, our focus is on computing equilibria. To address approximate generalized equilibria, we adopt the following definition from Rosen (1965).

**Definition 1 (Approximate Generalized Nash Equilibrium (GNE))**
*We call $\bar{x} = [\bar{x}^{(1)}, \ldots, \bar{x}^{(n)}] \in \mathbb{R}^{d^{([n])}}$ with $\bar{x}^{(i)} \in \mathbb{R}^{d^{(i)}}$ an $\epsilon$–approximate Generalized Nash Equilibrium if for all $i \in [n]$:*

$$\nu^{(i)}\left(x^{(i)}, \bar{x}^{(-i)}\right) \geqslant \nu^{(i)}(\bar{x}^{(i)}, \bar{x}^{(-i)}) - \epsilon \qquad \forall x \in \mathfrak{X}^{(i)}(\bar{x}^{(-i)}).$$

*If this inequality holds for $\epsilon = 0$, we call $\bar{x}$ a Generalized Nash Equilibrium.*

Due to Theorem 1 in Rosen (1965), Assumption 2.1 ensures the existence of a GNE. Note that computing the GNE reduces to standard Nash-equilibrium computation if for all $i \in [n]$ the constraints are independent of the opponents choice, that is $\mathfrak{X}^{(i)}(\bar{x}^{(-i)}) = \mathfrak{X}^{(i)}$ for all $x^{(-i)}$. Then $\mathfrak{C}$ reduces to the product space $\prod_{i \in [n]} \mathfrak{X}^{(i)}$. We call any constraint set $\mathfrak{C}$ which can be defined as a product space *uncoupled*, conversely, if this does not hold, we call it *coupled*.

**Benign GNEP.** Let $\phi > 0$. To define the classes of benign and strongly benign GNEP, we define for any $(x^{(i)}, x^{(-i)})$ the set

$$\hat{\mathrm{B}}_2^{(i)}(x^{(i)}, \phi) := \mathrm{B}_2^{(i)}(x^{(i)}, \phi) \cap \mathcal{X}^{(i)}(x^{(-i)}),$$

that is, $\hat{\mathrm{B}}_2(x^{(i)}, \phi)$ gives the $\phi$-neighborhood around $x^{(i)}$, intersected with the set of feasible actions[1]. Based on this set, we define the parameter $D_{\min}^{(i)}(\phi)$ as

$$D_{\min}^{(i)}(\phi) := \max \left\{ \alpha \mid \forall x \in \mathfrak{C}, \forall x_+^{(i)} \in \hat{\mathrm{B}}_2(x^{(i)}, \phi) + \left\{ -\alpha \frac{\nabla_i \nu^{(i)}(x^{(i)}, x^{(-i)})}{\|\nabla_i \nu^{(i)}(x^{(i)}, x^{(-i)})\|} \right\} : \left( x_+^{(i)}, x^{(-i)} \right) \in \mathfrak{C} \right\}$$

and define $D_{\min}(\phi) = \min_{i \in [n]} D_{\min}^{(i)}(\phi)$. The interpretation of $D_{\min}(\phi)$ is that it gives us the maximal quantity for which the $\phi$-neighborhood around any point in $\mathfrak{C}$ can be shifted in the direction of the (normalized) gradients and still stays inside $\mathfrak{C}$. Based on these definitions, we introduce the class of strongly benign and benign GNEP.

**Definition 2 (Strongly Benign GNEP)** *For $\phi, \delta > 0$, we call a GNEP $(\phi, \delta)$-strongly benign if*

1. *The GNEP has a unique GNE $u = (u^{(1)}, \ldots, u^{(n)}) \in \operatorname{relint} \mathfrak{C}$ such that for any $i \in [n]$ and $x^{(-i)} \in \mathcal{X}^{(-i)}$, we have $\|\nabla_i \nu^{(i)}(u^{(i)}, x^{(-i)})\| = 0$;*

2. *For all players $i \in [n]$ and all $(x^{(i)}, x^{(-i)}) \in \mathfrak{C}$ with $x \neq u$, it holds that $\nabla_i \nu^{(i)}(x^{(i)}, x^{(-i)})$ is sufficiently aligned with the vector $x^{(i)} - u^{(i)}$, i.e., there exists $\delta > 0$ such that*

$$\left\langle \nabla_i \nu^{(i)}(x^{(i)}, x^{(-i)}), x^{(i)} - u^{(i)} \right\rangle \geq \delta \left\| \nabla_i \nu^{(i)}(x^{(i)}, x^{(-i)}) \right\| \left\| x^{(i)} - u^{(i)} \right\|;$$

3. $D_{\min}(\phi) > 0$.

Note that for $\left\| \nabla_i \nu^{(i)}(x^{(i)}, x^{(-i)}) \right\| \neq 0$ and $x^{(i)} \neq u^{(i)}$, we can rewrite Condition 2 as

$$\frac{\left\langle \nabla_i \nu^{(i)}(x^{(i)}, x^{(-i)}), x^{(i)} - u^{(i)} \right\rangle}{\left\| \nabla_i \nu^{(i)}(x^{(i)}, x^{(-i)}) \right\| \left\| x^{(i)} - u^{(i)} \right\|} \geq \delta,$$

that is, we rewrite it as an angular condition on the vectors $\nabla_i \nu^{(i)}(x^{(i)}, x^{(-i)})$ and $x^{(i)} - u^{(i)}$. Therefore, we will refer to it as the *benign angular condition*.

**Definition 3 (Benign GNEP)** *For $\phi, \delta, \Delta > 0$, we call a GNEP $(\phi, \delta, \Delta)$-benign if it satisfies Conditions 2 and 3 from Definition 2, and, in addition,*

1. *The GNEP has a unique GNE $u = (u^{(1)}, \ldots, u^{(n)}) \in \operatorname{relint} \mathfrak{C}$ such that for any $i \in [n]$ and $x^{(-i)} \in \mathcal{X}^{(-i)}$, we have $\left\| \nabla_i \nu^{(i)}(x^{(i)}, x^{(-i)}) \right\| \geq \Delta \left\| x^{(i)} - u^{(i)} \right\|$.*

---

1. Recall from Section 1 that $\mathrm{B}_2(x, \epsilon)$ defines the $\ell_2$ ball around the point $x$ with radius $\epsilon$.

### 2.2. Relation to Common Existing Assumptions

In this section, we set our assumption in context with existing common assumptions. First, note that under Assumption 2.1, any $(\phi, \delta)$-strongly benign GNEP is $(\phi, \delta, \Delta)$-benign with $\Delta = \mu$. To see this, note that due to Assumption 2.1, the utilities are $\mu$-strongly convex. Hence, for any strongly benign GNEP

$$\left\langle \nabla_i \nu^{(i)}\big(x^{(i)}, x^{(-i)}\big) - \nabla_i \nu^{(i)}\big(u^{(i)}, x^{(-i)}\big), x^{(i)} - u^{(i)} \right\rangle \geqslant \mu \left\| x^{(i)} - u^{(i)} \right\|^2$$

$$\Rightarrow \left\| \nabla_i \nu^{(i)}\big(x^{(i)}, x^{(-i)}\big) - \nabla_i \nu^{(i)}\big(u^{(i)}, x^{(-i)}\big) \right\| \left\| x^{(i)} - u^{(i)} \right\| \geqslant \mu \left\| x^{(i)} - u^{(i)} \right\|^2.$$

Reordering the terms and using that for strongly benign GNEP $\left\| \nabla_i \nu^{(i)}\big(u^{(i)}, x^{(-i)}\big) \right\| = 0$ gives the claim. To introduce further relations, we recall the definition of quasi-variational inequalities.

**Definition 4 (Quasi Variational Inequality)** *Let $F : \mathbb{R}^d \to \mathbb{R}^d$ and consider a set valued mapping $C : \mathbb{R}^d \to 2^{\mathbb{R}^d}$. For a quasi-variational inequality problem we want to find $x^\star \in C(x^\star)$ such that*

$$\langle F(x^\star), x^\star - y \rangle \geqslant 0 \qquad \forall y \in C(x^\star).$$

*We denote this problem by* $\mathrm{QVI}(F, C)$

Suppose $\mathfrak{C}$ is closed and convex and for all $x^{(-i)} \in \mathbb{R}^{d^{(-i)}}$, $\nu^{(i)}(\,\cdot\,, x^{(-i)})$ is convex, continuous and differentiable, then computing a GNE is equivalent to solving a quasi-variational inequality (QVI) with $F(x) = [\nabla \nu^{(1)}(\,\cdot\,, x^{(-1)}), \dots, \nabla \nu^{(n)}(\,\cdot\,, x^{(-n)})]$ and $C(x) = \prod_{i \in [n]} \mathfrak{X}^{(i)}(x^{(-i)})$. This relation was first discovered by Bensoussan (1974). For more details see for example Section 2 in Pang and Fukushima (2005) or Theorem 2 in Facchinei and Kanzow (2007).

Consider a convex set $\mathfrak{C}$ and suppose for all $x \in \mathfrak{C}$, $C(x) = \mathfrak{C}$, then $\mathrm{QVI}(F, C)$ reduces to a variational inequality problem $\mathrm{VI}(F, \mathfrak{C})$:

$$\text{find } x^\star \in \mathfrak{C}, \text{ such that} \quad \langle F(x^\star), x^\star - y \rangle \geqslant 0 \qquad \forall y \in \mathfrak{C}.$$

Note that solving a standard Nash-equilibrium problem (NEP) without shared constraints is equivalent to solving a variational inequality problem. Hence, assuming that $C(x) = \mathfrak{C}$, reduces the GNEP to a standard NEP. However, there exist sufficient conditions under which the solutions of a QVI are equal to the solutions of a VI. One such condition is due to a result by Harker (1991): If $A \subset \mathfrak{C}$ such that $\forall x \in A, x \in C(x) \subset A$, then any solution to the $\mathrm{VI}(F, A)$ is a solution to the $\mathrm{QVI}(F, C)$. However, as noted by Facchinei et al. (2007), in the case of a QVI arising from a GNEP, this condition only allows for uncoupled sets. Hence, under this condition, the problem reduces to an NEP. Note that the strongly benign condition allows for problems that do not satisfy Harker's condition. (See Section 2.3 for examples and more details.)

A common assumption for variational inequalities is the strong monotonicity assumption: An operator $F : \mathfrak{X} \to \mathfrak{X}$ is $\mu$-strongly monotone if $\forall x, y \in \mathfrak{X}$

$$\langle F(x) - F(y), x - y \rangle \geqslant \mu \|x - y\|^2,$$

and strictly monotone if $\langle F(x) - F(y), x - y \rangle > 0$. Strong and strict monotonicity are common assumptions in the context of monotone operator theory or variational inequalities.[2] In the context of QVI, strong monotonicity is considered in Nesterov and Scrimali (2006). In the special

---

2. In the context of game theory, strict monotonicity is also referred to as *diagonal strict concavity* (Rosen, 1965).

case of games, strong monotonicity is for example considered in Duvocelle et al. (2023); Yan et al. (2023) and referred to as *strongly monotone games*. In the case of $\mu$-strongly convex loss functions (cf. Assumption 2.1, 2.), the game is $\mu$-strongly monotone. We note that under the additional assumption that $\nabla_i \nu^{(i)}(\,\cdot\,, x^{-i})$ are injective and their inverse is $L^{-1}$-Lipschitz continuous (a.k.a. $L$-bi-Lipschitz), $(\phi, \delta)$-strongly benign GNEP are $\frac{\delta}{L}$-strongly monotone. Conversely, if the loss functions are $\mu$-strongly convex and $L$-smooth and for GNE $u \in \mathfrak{C}$, $\nabla_i \nu^{(i)}(u^{(i)}, x^{(-i)}) = 0$ then the benign angular condition is satisfied. This is formalized by the following Proposition:

**Proposition 5**

1. *Consider a GNEP with $\mu$-strongly convex loss functions. Further, suppose $\nu^{(i)}(\,\cdot\,, x^{-i})$ are $L$-smooth and there exists a GNE $u \in \mathfrak{C}$ such that $\nabla_i \nu^{(i)}(u^{(i)}, x^{(-i)}) = 0$ for all $x \in \mathfrak{C}$. Then the benign angular condition is satisfied with $\delta = \mu L$.*

2. *Consider a strongly benign GNEP (cf. Definition 2) and assume that the $\nabla_i \nu^{(i)}(\,\cdot\,, x^{-i})$ are $L$-bi-Lipschitz. Then the game is $(\delta/L)$-strongly monotone.*

   For a proof, see Appendix A.

### 2.3. Examples of Strongly Benign GNEP

The conditions for benign and strongly benign GNEP are strong and exclude many instances of GNEP. However, it is a nontrivial subclass of GNEP as we illustrate in the following examples. The first example serves as an illustration of the definitions and is relatively trivial. The reason is that the example is one-dimensional. This enables us to illustrate it via plots (see Section 3.2). However, conditions such as the benign angular conditions constrain us to trivial uncoupled loss functions in one dimension.

**Example 1 ($(\phi, \delta)$-strongly benign GNEP)** *Consider a two-player game where $d^{(1)} = d^{(2)} = 1$. For simplicity, we denote the players action by $x \in \mathbb{R}$ and $y \in \mathbb{R}$ instead of $x^{(1)}$ and $x^{(2)}$. Furthermore, we denote the sets $\mathcal{S}_t^{(1)}, \mathcal{S}_t^{(2)}$ by $\mathcal{S}_t^{(x)}, \mathcal{S}_t^{(y)}$. Define the shared constraint set*

$$\mathfrak{C} = \{(x, y) \in \mathbb{R}^2 : 8 \geqslant x \geqslant -1, \ 8 \geqslant y \geqslant -1, \ x + y \leqslant 10\}.$$

*The players want to minimize their losses with respect to $\mathfrak{C}$. That is*

$$\min_{x \in \mathbb{R}} \nu^{(x)}(x, y) \quad s.t. \quad (x, y) \in \mathfrak{C}$$
$$\min_{y \in \mathbb{R}} \nu^{(y)}(y, x) \quad s.t. \quad (x, y) \in \mathfrak{C}\,.$$

*The loss function for the $x$-player is $\nu^{(x)}(x, y) = x^2 - y^2$ and for the $y$-player it is $\nu^{(y)}(y, x) = y^2 - x^2$.*

*We show that this problem is $(1 - \epsilon, 1)$-strongly benign for any $\epsilon > 0$. To avoid repetitive arguments, we only verify all conditions for the $x$-player. Due to the symmetry of the problem, the same argument can be applied to the $y$-player.*

*First note that the unique GNE is $[u^{(x)}, u^{(y)}] = [0, 0] \in \mathfrak{C}$ and the gradient norm $\nabla_x \nu^{(x)}(0, y) = 0$ for any choice of $y \in \mathcal{X}^{(y)}(0)$. Hence, condition (1) of Definition 2 is satisfied. Further, the benign angular condition is satisfied with $\delta = 1$ since*

$$\frac{\langle \nabla_x \nu^{(x)}(x, y), x - u^{(x)} \rangle}{\left\| \nabla_x \nu^{(x)}(x, y) \right\| \left\| x - u^{(x)} \right\|} = \frac{\langle 2x, x \rangle}{\left\| 2x \right\| \left\| x \right\|} = 1.$$

*Hence, the benign angular condition in Definition 2 is satisfied. To verify that $D^{(x)}_{\min}(1 - \epsilon) > 0$, note that $\frac{\nabla_x \nu^{(x)}(x,y)}{\|\nabla_x \nu^{(x)}(x,y)\|} = \frac{2x}{|2x|}$. Further, $\frac{2x}{|2x|}$ is equal to the sign of $x$, that is $\frac{2x}{|2x|} = -1$ if $x < 0$ and $\frac{2x}{|2x|} = 1$ for $x > 0$ and $\frac{2x}{|2x|} = 0$ for $x = 0$. We denote this by $\mathrm{sign}(x)$. Consider any $[x,y] \in \mathfrak{C}$. We verify that $\mathrm{relint}\, \mathfrak{X}^{(x)}(y) \ni 0$. Moreover, the distance between $\mathrm{bd}\, \mathfrak{X}^{(x)}(y)$ and $0$ is always a constant. Together with the observation that $\frac{2x}{|2x|} = \mathrm{sign}(x)$, this implies that $D_{\min}(1 - \epsilon) > 0$ for any $\epsilon > 0$. Hence, we conclude that the problem is $(1 - \epsilon, 1)$-strongly benign. We note that Harker's condition is not satisfied for this example since the constraints are coupled.*

The following example is an illustration of a strongly benign GNEP beyond one dimension.

**Example 2 ($(\phi, \delta)$-strongly benign GNEP)** *We consider a two-player GNEP and use the same notation simplification as in Example 1.*

$$\min_{x \in \mathbb{R}^2} \max_{y \in \mathbb{R}^2} \|x\|^2 - \|y\|^2 + \left(x^\top P y\right)^2 \quad s.t. \quad (x,y) \in \mathfrak{C}.$$

*Where $P \in \mathbb{R}^{2 \times 2}$ is a positive definite matrix and for $\epsilon > 0$*

$$\mathfrak{C} = \{(x,y) \in \mathbb{R}^{2 \times 2} : x \in [-1,1]^2,\ y \in [-1,1]^2, x_1 + y_1 + \epsilon x_2 + \epsilon y_2 \leqslant 1\}.$$

*We note that the unique GNE is at $(0,0)$. Similar to the previous example, we can check that the conditions for strongly benign GNEP are satisfied.*

## 3. Online Feasible Point Method: Algorithm

In this section, we introduce an online FPM with alternating coordination: the players coordinate by making sure that they update *their desired set* every $n$ turns. The goal of this algorithm is to guarantee convergence to the GNE while preserving feasibility for all iterates. In Section 4, we show that this is guaranteed for an interesting class of games under the assumption that all players use the introduced algorithm. Throughout this section, we assume that Assumption 2.1 is satisfied. Recall that the parameter $G$ denotes an upper bound on the gradient norm of the loss functions. Further, we let $D = \mathrm{Diameter}(\mathfrak{C})$ and assume $D$ is known to the players.

**A Naive Algorithm: Waiting for Your Turn.** A first method that fits our framework is to have the players update their iterates one player at a time, with every player using an optimization method such as (projected) gradient descent. Formally, player $i$, sends the desired set $\mathcal{S}^{(i)}_t = \{x^{(i)}_t\}$ at every turn, and changes the iterate $x^{(i)}_t$ only at timesteps $i + kn$ for $k \geqslant 0$.

While this method technically fits our framework, having the players wait for their turn is wasteful as it does not profit from the fact that players may update their plays more often if they declare larger sets $\mathcal{S}^{(i)}_t$. In the rest of this paper, we design and analyze a full algorithm in which the players update they play simultaneously while maintaining the joint feasibility thanks to the prior declaration of the desired sets.

### 3.1. Online Feasible Point Method with Alternating Coordination

Throughout this section, we assume that all players have access to first-order information, i.e., to gradients $g^{(i)}_t = \nabla_i \nu^{(i)}(x^{(i)}_t, x^{(-i)}_t)$. Furthermore, to guarantee feasibility, the players define

and communicate the desired sets $\mathcal{S}_{t+1}^{(i)}$ at the end of each round $t$. In the next round, all players $i \in [n] \setminus \{j\}$ choose their iterate $x_{t+1}^{(i)}$ from $\mathcal{S}_{t+1}^{(i)}$. The $j^{th}$ player is allowed to update the set $\mathcal{S}_{t+1}^{(j)}$ and chooses its iterate from this set. To guarantee that all players can make sufficient progress, the player who moves its set varies every iteration. We assume that for each player $i \in [n]$, the online FPM is initialized with $x_1^{(i)}$, and closed and bounded convex set $\mathcal{S}_1^{(i)}$ such that $x_1^{(i)} \in \operatorname{relint} \mathcal{S}_1^{(i)}$ and $\mathcal{S}_1^{\times [n]} \subset \mathfrak{C}$.

---

**Algorithm 1** Online Feasible Point Method with Alternating Coordination

---

**Input:** Convex set $\mathcal{S}_1^{(i)} \subset \mathbb{R}^{d_i}, x_1^{(i)} \in \operatorname{relint} \mathcal{S}_1^{(i)}$
set $p_{\text{start}}(1) = 1$ and $k = 1$
**for** $t = 1, \ldots, T$ **do**
$\quad$ **if** *Termination Criterion* (**TC**) *not satisfied* **then**
$\quad\quad$ play $x_t^{(i)}$, suffer loss and receive gradient $g_t^{(i)}$
$\quad\quad$ update $x_{t+1}^{(i)}$ according to (**Update**)
$\quad\quad$ update $\mathcal{S}_{t+1}^{(i)}$ according to (**Vupdate**)
$\quad\quad$ send $\mathcal{S}_{t+1}^{(i)}$ to all players $j \in [n] \setminus \{i\}$ and receive $\mathcal{S}_{t+1}^{(j)}$ from all $j \in [n] \setminus \{i\}$
$\quad$ **end**
$\quad$ **if** *Termination Criterion* (**TC**) *satisfied* **then**
$\quad\quad$ set $x_{t+1}^{(i)} = x_t^{(i)}$
$\quad\quad$ shrink $\mathcal{S}_t^{(i)}$ such that $x_t^{(i)} \in \operatorname{relint} \mathcal{S}_{t+1}^{(i)}$ and $\operatorname{Diameter}\big(\mathcal{S}_{t+1}^{(i)}\big) = \frac{1}{2}\operatorname{Diameter}\big(\mathcal{S}_t^{(i)}\big)$
$\quad\quad$ set $p_{\text{end}}(k) = t$ and $p_{\text{start}}(k+1) = t+1$ and increment $k$ by one
$\quad$ **end**
**end**

---

The online FPM proceeds in phases. The $k^{\text{th}}$-phase starts at $p_{\text{start}}(k)$ and ends at $p_{\text{end}}(k)$. We note that $p_{\text{start}}(k) \leqslant p_{\text{end}}(k)$ and denote $\mathscr{P}_k = \{p_{\text{start}}(k), \ldots, p_{\text{end}}(k)\}$. It remains to specify the termination criterion (**TC**) and the updates of the iterates $x_t^{(i)}$ and $\mathcal{S}_t^{(i)}$, (**Update**) and (**Vupdate**) respectively. The termination criterion is triggered if none of the players can make sufficient progress or if it was not triggered for $2^k$ iterations. That is,

$$\max_{i \in [n]} \operatorname{dist}\left(\mathcal{S}_{t-i}^{[\times[n]]}, \mathcal{S}_{t-i+1}^{[\times[n]]}\right) \leqslant \frac{D}{\sqrt{T}} \qquad \text{or} \qquad t - p_{\text{start}}(k) \geqslant 2^k. \tag{TC}$$

To define the updates of $x_t^{(i)}$ and $\mathcal{S}_t^{(i)}$, we let $\eta > 0$ denote a fixed step size to be defined later. The updates are done with respect to the minimum between $\eta$ and the parameters $\iota_t \geqslant 0$ and $\bar{\eta}_t/2 > 0$ to guarantee feasibility. To define $\iota_t$, set $\mathcal{Z}_t^{(i)} = \{z : z \in \operatorname{argmin}_{z \in \mathcal{S}_t^{(i)}} \langle g_t^{(i)}, z \rangle\}$. Define

$$\iota_t^{(i)} = \min \operatorname{dist}(\mathcal{Z}_t^{(i)} \times \mathcal{S}_t^{(-i)}, \operatorname{bd}(\mathfrak{C})).$$

That is, $\iota_t^{(i)}$ is the minimal distance with respect to the gradient direction $g_t^{(i)}$ of the set $\mathcal{S}_t^{\times [n]}$ to the boundary of $\mathfrak{C}$. For the updates of the set $\mathcal{S}_t^{(i)}$, we set

$$\mathcal{S}_{t+1}^{(i)} = \begin{cases} \mathcal{S}_t^{(i)} & \text{if } t \bmod n \neq i - 1, \\ \mathcal{S}_t^{(i)} + \{v_t^{(i)}\} & \text{if } t \bmod n = i - 1 \end{cases}, \tag{Vupdate}$$

where $v_t^{(i)} = -\min(\eta, \iota_t^{(i)})g_t^{(i)}$. Further, we set $\bar{\eta}_t^{(i)} = \max(\alpha \geqslant 0 : (x_t^{(i)} - \alpha g_t^{(i)}) \in \mathcal{S}_t^{(i)})$, that is, $\bar{\eta}_t^{(i)}$ is the maximal step into the direction of $g_t^{(i)}$ while staying within $\mathcal{S}_t^{(i)}$. Define the update

$$x_{t+1}^{(i)} = \begin{cases} x_t^{(i)} - \min(\eta, \bar{\eta}_t^{(i)}/2)\, g_t^{(i)} & \text{if } t \bmod n \neq i - 1 \\ x_t^{(i)} + v_t^{(i)} & \text{if } t \bmod n = i - 1 \end{cases}, \qquad \textbf{(Update)}$$

where $v_t^{(i)}$ is defined as before.

**Remark 6 (Computational complexity)** *The computational complexity of each iteration depends on computing $\iota_t^{(i)}$, which requires evaluating $\min \mathrm{dist}(\mathfrak{X}_t^{(i)} \times \mathcal{S}_t^{(-i)}, \mathrm{bd}(\mathfrak{C}))$. The shared constraint set $\mathfrak{C}$ is problem-dependent and typically defined via $k \in \mathbb{N}$ convex functions $h_i : \mathbb{R}^{d^{[n]}} \to \mathbb{R}$ as $\mathfrak{C} = \{x \in \mathbb{R}^{d^{[n]}} : \forall i \in [k] : h_i(x^{(1)}, \dots, x^{(n)}) \leqslant 0\}$. In contrast, $\mathcal{S}_t^{(i)}$ is part of the algorithm definition. While we do not impose any assumptions on $\mathcal{S}_t^{(i)}$ beyond convexity, it is beneficial for computational efficiency to define it as a convex polytope with a small number of vertices, e.g., set $\mathcal{S}_1^{(i)}$ as a $d^{(i)}$-dimensional nonstandard simplex. In this case, $\mathcal{S}_t^{(i)} \times \mathcal{S}_t^{(-i)}$ contains $d^{[n]} + n$ vertices and hence $\iota_t^{(i)}$ can be computed via $(d^{[n]} + n)k$ line searches. In particular, if the functions $h_i$ are affine, i.e., $\mathfrak{C}$ is also a polytope, then the computation of $\iota_t^{(i)}$ requires solving $(d^{[n]} + n)k$ linear equations. In both cases, the computational complexity is polynomial in the number of edges of $\mathcal{S}_t^{(i)} \times \mathcal{S}_t^{(-i)}$.*

### 3.2. Illustrating Examples for the Alternating Coordination online FPM

To provide a better intuition for our algorithm, we illustrate its computations with an example.

**Example 3 (Example 1 continued)** *In the following we continue with Example 1. Figure 1 shows the execution of the first steps for the online FPM defined in Algorithm 1. The initial iterates are $x_1 = 2, y_1 = 6$ and $\mathcal{S}_1^{(x)} = [1, 3]$, $\mathcal{S}_1^{(y)} = [5, 7]$. We note that $x_1 \in \mathcal{S}_1^{(x)}$ and $y_1 \in \mathcal{S}_1^{(y)}$. Further, $\mathcal{S}_1^{(x)} \times \mathcal{S}_1^{(y)} \subseteq \mathfrak{C}$. For better visualization, we only plot every second step. The red dot indicates the combined players' iterates, i.e., the vector $[x_t, y_t]$. The boxes denote the product sets $\mathcal{S}_t^{(x)} \times \mathcal{S}_t^{(y)}$. The color highlights the algorithm's progress: the first step is in dark red, and the last iterate in dark blue.*

## 4. Online Feasible Point Method: Feasibility and Convergence

In this section, we show that the online FPM is guaranteed to preserve feasibility. Furthermore, we show that for strongly being GNEP with $L$-smooth and $\mu$-strongly convex loss functions (cf. Assumption 2.1), convergence to the GNE is guaranteed.

**Theorem 7** [Feasibility] *Suppose all players are following Algorithm 1. Then for all iterations $t \in [T]$, we have $\mathcal{S}_t^{[\times[n]]} \subseteq \mathfrak{C}$; this implies in particular that $[x_t^{(1)}, \dots, x_t^{(n)}] \in \mathfrak{C}$.*

This result is a straightforward consequence of the definition of the algorithms, for a formal proof see Appendix C. Furthermore, for strongly benign and benign GNEP, the online FPM converges to the unique GNE.

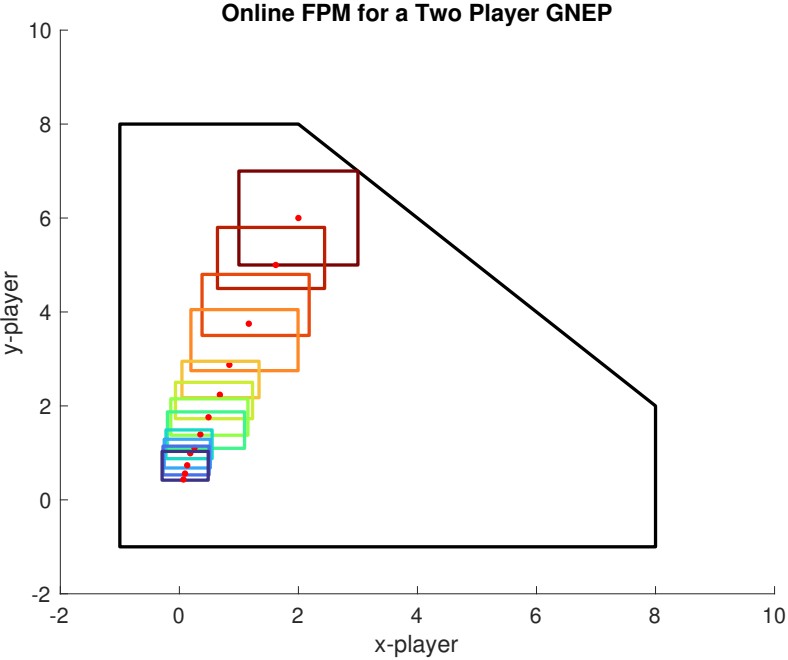

Figure 1: The first 24 iterations of the online FPM Algorithm 1 on the GNEP defined in Example 1.

**Theorem 8** *[Convergence] Suppose Assumption 2.1 is satisfied and assume all players are follow-ing Algorithm 1 with step size $\eta = \min(\frac{D}{G\sqrt{T}}, \frac{\delta}{L})$. Assume we have a $(\phi, \delta)$-strongly benign GNEP. Set*

$$t_0 = \max\left(\frac{4D}{\phi} + 1, \left(\frac{D}{2D_{\min}}\right)^2, \left(\frac{DL}{2G\delta}\right)^2\right).$$

*Then for all players $i \in [n]$ and any $t \in [t_0, T]$*

$$\left\|x_t^{(i)} - u^{(i)}\right\| \leqslant \Xi \left\|x_1^{(i)} - u^{(i)}\right\| \left(1 - \frac{\mu\delta D}{4G\sqrt{T}}\right)^{\frac{t+1}{n}} + \frac{2D}{\delta\sqrt{T}},$$

*where $\Xi = \left(1 - \frac{\mu\delta D}{4G\sqrt{T}}\right)^{-\frac{t_0}{n}}$.*

For a complete proof and a similar result for benign GNEP, see Appendix D. Our proof relies on a special case of an inexact gradient descent. That is, a gradient descent scheme of the form $x_{t+1} = x_t - \eta_t g_t$ where $g_t$ does not necessarily correspond to the gradient, but is a vector that is sufficiently aligned with the real gradient. Hence, the benign angular condition is essential for this analysis since it ensures that the gradients of the loss function are relatively well aligned with the gradients of the function $f(x) = 2\delta^{-1}\|x - u\|$. See Section D.1 in the Appendix for more details and Remark 17 for comments on related results.

### 4.1. Comparison to Alternating Gradient Descent

As mentioned in Section 3, an alternative approach to the online FPM (Algorithm 1) would be an alternating gradient descent method. That is, for each step $t \in [T]$, each player $i \in [n]$ sets

$$x_{t+1}^{(i)} = \begin{cases} x_t^{(i)} - \eta_t g_t^{(i)} & \text{if } t \bmod n = i - 1 \\ x_t^{(i)} & \text{otherwise.} \end{cases} \tag{Alt-GD}$$

This method has the clear advantage of being easier to implement and requiring no extra storage for the sets $\mathcal{S}_t^{(i)}$. Note that the feasibility of the iterates can be guaranteed if the parameter $D_{\min}(0)$ is available for stepsize tuning or all $x_t^{(j)}$ of all players $j \in [n] \setminus \{i\}$ are known to player $i \in [n]$. Indeed, in the latter case, feasibility can be guaranteed by a projection onto $\mathcal{X}^{(i)}(x_t^{(-i)})$ and in the former case, feasibility with $\eta = D_{\min}(0)/(\sqrt{T}G)$ follows from the definition. Thus, assuming that either of these parameters is available can also guarantee feasibility for this algorithm. However, note that alternation is essential for this argument; therefore, a player can only make progress every $n^{th}$-iteration. Conversely, note that with the online FPM, Algorithm 1, the players update their iterates every iteration while guaranteeing feasibility. While this does not always help, in some cases of practical interest it can lead to a significant speedup as the following example shows:

**Example 4** *Consider a GNEP with $\mathfrak{C} = \prod_{i \in [n]} \Delta_d$ and $\nu^{(i)}\big(x^{(i)}, x^{(-i)}\big)$ such that there exists a unique GNE $(\bar{x}^{(1)}, \ldots, \bar{x}^{(n)})$ in the strict interior of $\mathfrak{C}$. Comparing the online FPM (Algorithm 1) with the alternating gradient descent method (Alt-GD), we note that the players can update their iterates in the online FPM every iteration, while for (Alt-GD), they only make progress every $n^{th}$ iteration. Furthermore, for this specific problem, given that $\bar{x} \in \text{int}\left(\mathcal{S}_t^{(\times[n])}\right)$ every player makes progress towards the GNE with a step size of at least $\min(\text{Diameter}\big(\mathcal{S}_t^{(i)}\big)/(2G), \eta)$. Given that $\text{Diameter}\big(\mathcal{S}_t^{(i)}\big)$ is a constant, each player makes constant progress towards the GNE. Specifically in the case when the number of players is large, e.g., proportional to $\sqrt{T}$, this speed-up can be significant and of practical interest.*

Another advantage of the online FPM is its robustness beyond the theoretical guarantees. As our experiments show (cf. Appendix B), the online FPM can work well beyond the theoretical guarantees. In particular, it always guarantees feasibility and convergence to GNE for instances beyond the class of benign GNEP. Capturing the described advantages of the online FPM in a rigorous analysis constitutes an interesting future research direction (see Section 6 for more details).

## 5. Relation to Regret Minimization

In this section, we highlight the relation of our algorithm to no-regret algorithms in repeated games. From the perspective of a single player, interacting in a game with shared constraints can be viewed as a convex online learning problem with time-varying constraints. Note that in contrast to standard online learning problems, where the constraint is fixed, online learning with time-varying constraint sets is a fundamentally more challenging setting. In recent years this setting received more interest in the online learning community (see e.g. Neely and Yu (2017); Liu et al. (2021); Kolev et al. (2023)).

Consider the following online learning protocol with time-varying constraints: For each round $t \in [T]$, the learner receives side information $\hat{\mathscr{C}}_t \subset \mathcal{X}$ and chooses an iterate $x_t \in \mathcal{X}$. Next, she

receives a convex loss function $f_t : \mathcal{X} \to \mathbb{R}$ and a convex constraint set $\mathcal{C}_t = \{x \in \mathcal{X} \mid h_t(x) \leqslant 0\} \subset \mathcal{X}$ with $h_t : \mathcal{X} \to \mathbb{R}$ and suffers instantaneous loss with respect to $f_t(x_t)$ plus a cost for violating the constraint $\mathcal{C}_t$. A common evaluation criterion for the performance of an algorithm for an online learning problem is the *regret*. For any $u \in \mathcal{X}$, define

$$\mathrm{Reg}_T^f(u) := \sum_{t=1}^{T} f_t(x_t) - \sum_{t=1}^{T} f_t(u). \tag{1}$$

An online algorithm is said to be a *no-regret algorithm* if for any $u \in \mathcal{X}$, $\mathrm{Reg}_T^f(u)$ grows sublinearly in $T$. Since the algorithm faces time-varying constraints, we also want to measure the performance with respect to the constraint violations. There exist several definitions of regret for constraint violation in the literature. For example, Mahdavi et al. (2012) use the regret definition $\sum_{t=1}^{T}(h_t(x_t) - h_t(u))$ and Yuan and Lamperski (2018) define the stronger version $\sum_{t=1}^{T}(h_t(x_t))_+ - (h_t(u))_+$ [3] to measure the constraint violations. There also exist last step requirement like $h_T(x_T) \leqslant O(1/\sqrt{T})$ as used in Kolev et al. (2023).

To capture the requirement of feasibility of every iterate, we introduce a definition of regret, which is based on the indicator function. That is, let $\chi_{\mathcal{C}_t} : \mathcal{X} \to \mathbb{R} \cup \{+\infty\}$ denote the characteristic function with $\chi_{\mathcal{C}_t}(x) = 0$ if $x \in \mathcal{C}_t$ and $+\infty$ otherwise. Define the regret with respect to constraint violations

$$\mathrm{Reg}_T^c = \sum_{t=1}^{T} \chi_{\mathcal{C}_t}(x_t). \tag{2}$$

Note that $\mathrm{Reg}_T^c$ is equal to the *dynamic regret* $\sum_{t=1}^{T} \chi_{\mathcal{C}_t}(x_t) - \chi_{\mathcal{C}_t}(u_t)$ where $u_t \in \mathcal{C}_t$. Furthermore, $\mathrm{Reg}_T^c \in \{0, +\infty\}$. Hence, requiring an algorithm to be a no-regret algorithm with respect to $\mathrm{Reg}_T^c$ is equivalent to requiring $\mathrm{Reg}_T^c = 0$. We note that for any sequence $(x_t)_{t \in [T]} \in \mathcal{X}^T$ and $u \in \mathcal{X}$

$$\sum_{t=1}^{T} \chi_{\mathcal{C}_t}(x_t) \geqslant \sum_{t=1}^{T} (h_t(x_t))_+ - (h_t(u))_+.$$

Hence, if an algorithm is no-regret with respect to $\mathrm{Reg}_T^c$, then $\sum_{t=1}^{T}(h_t(x_t))_+ - (h_t(u))_+$ is nonpositive. Note that the converse is not necessarily true. Furthermore, if $\mathrm{Reg}_T^c = 0$, this immediately implies that $h_t(x_t) \leqslant 0$ for all $t \in [T]$. Hence, we immediately obtain the last step requirement $h_T(x_T) \leqslant c\frac{1}{\sqrt{T}}$ from Kolev et al. (2023).

From Theorem 8 we derive sublinear regret bounds. We adapt the notation to be consistent with the previous notation: the convex loss functions are $\nu^{(i)}(\cdot, x_t^{(-i)})$, the constraint sets are $\mathcal{X}^{(i)}(x_t^{(-i)})$ and the side information the learner receives is $\mathcal{S}_t^{(-i)}$. Note that $\nu^{(i)}(\cdot, x_t^{(-i)})$ and $\mathcal{X}^{(i)}(x_t^{(-i)})$ are time-varying due to the choice of $x_t^{(-i)}$.

**Theorem 9** *Suppose Assumption 2.1 is satisfied and assume we have a $(\phi, \delta)$-strongly benign GNEP. Let $\Xi$ and $t_0$ be defined as in Theorem 8. If all players are following Algorithm 1, then for*

---

3. Note that if $\varnothing \neq \cap_{t \in [T]} \mathcal{C}_t \ni u$, this reduces to $\sum_{t=1}^{T}(h_t(x_t))_+$.

*all players $i \in [n]$*

$$\sum_{t=1}^{T} \chi_{\mathfrak{X}^{(i)}(x_t^{(-i)})}\left(x_t^{(i)}\right) = 0,$$

*and for all $T \in \mathbb{N}$*

$$\sum_{t=1}^{T} \nu^{(i)}\left(x_t^{(i)}, x_t^{(-i)}\right) - \sum_{t=1}^{T} \nu^{(i)}\left(u^{(i)}, x_t^{(-i)}\right) \leqslant DG\left(\sqrt{T}\left(2\Xi n\frac{G}{\mu D} + \frac{2}{\delta}\right) + t_0\right).$$

This result is a direct consequence of the convexity of the losses and Theorem 8. For a proof, see Appendix E. The regret bound provides a limited robustness guarantee for strongly benign GNEP: Given all players follow Algorithm 1, then no player has the incentive to unilateral deviate from this strategy.

From a single-player perspective, the online learning problem with time-varying constraints induced by a GNEP is fundamentally easier than an online learning problem with adversarially time-varying constraints. The following three aspects are essential: (1) the constraints are not adversarial but endogenous to the system, (2) before committing to the next iterate, the player has access to side information $\mathcal{S}_t^{(i)} \subseteq \mathfrak{X}^{(i)}(x_t^{(-i)})$, and (3) the distance between $u^{(i)}$ and $\mathcal{S}_t^{(i)}$ is decreasing. We note that if the time-varying constraints are chosen adversarially, it is not possible for any online algorithm to guarantee that $\mathrm{Reg}_T^c$ is bounded[4]. Based on these insights, we deduce an online learning protocol where the learner receives a set $\mathcal{S}_t \subseteq \mathscr{C}_t$ as side information before he has to choose his next iterate. The protocol is as follows: In each round $t \in [T]$, the learner receives a set $\mathcal{S}_t \subseteq \mathscr{C}_t$ and then chooses the iterate $x_t$. Next, the player learns $f_t$ and $\mathscr{C}_t$ and suffers a loss with respect to $f_t$. If the player violates the constraint $\mathscr{C}_t$, he suffers a loss of $\infty$.

By choosing its iterates $x_t \in \mathcal{S}_t$, the learner can guarantee that the constraints $\mathscr{C}_t$ are never violated. However, it is not necessarily possible to simultaneously guarantee sub-linear regret for $\mathrm{Reg}_T^f(u)$. For illustration, we consider the following simple example: Suppose the relative interior of $\cap_{t=1}^{T} \mathscr{C}_t$ is non-empty and take any $x, y \in \cap_{t=1}^{T} \mathscr{C}_t$ with $\mathrm{dist}(x, y) > 0$ a constant. Suppose the learner is provided $\mathcal{S}_t = \{x\}$ for $t \leqslant T/2$ and $\mathcal{S}_t = \{y\}$ otherwise. Let $u \in \cap_{t=1}^{T} \mathscr{C}_t$ and define $f_t(x) = a_t \|x - u\|^2$ for any $a_t > 0$. Then $\mathrm{Reg}_T^f(u)$ is linear in $T$ whenever the player chooses $x_t \in \mathcal{S}_t$ to avoid constraint violations. Based on these insights, we derive the following result: Suppose the learner obtains sets $\mathcal{S}_t \neq \varnothing$ with $\mathcal{S}_t \subset \mathscr{C}_t$. Assume that the distance $\mathrm{dist}(\mathcal{S}_t, u) \leqslant \frac{c}{\sqrt{t}}$ and the distance between the sets $\mathcal{S}_t$ and $\mathcal{S}_{t+1}$, i.e., $\max_{a \in \mathcal{S}_t} \mathrm{dist}(\mathcal{S}_{t+1}, a)$, is uniformly bounded for all $t \in [T]$. Assume the functions $f_t : \mathbb{R}^d \to \mathbb{R}$ are convex and differentiable. Then the regret for projected online gradient descent, i.e., $x_{t+1} = \mathrm{Proj}_{\mathcal{S}_{t+1}}(x_t - \eta_t \nabla f_t(x_t))$, is sublinear while the constraints are never violated. That is,

$$\mathrm{Reg}_T^f(u) \leqslant C\frac{3D + 10c}{2}G\sqrt{T} \quad \text{and} \quad \sum_{t=1}^{T} \chi_{\mathscr{C}_t}(x_t) = 0,$$

where $D \geqslant \mathrm{Diameter}(\mathcal{S}_t)$, $G \geqslant \|\nabla f_t(x_t)\|$ and $C > 0$ denotes a constant independent of $T$. See Lemma 19 in Appendix E.1 for more details.

---

4. For example, consider $\mathfrak{X} = [-1, 1]$. For any sequence of $\{x_t\}_{t \in [T]}$ there exists a sequence of adversarial time-varying constraints $\{\mathscr{C}_t\}_{t \in [T]}$, $\mathscr{C}_t \subset \mathfrak{X}$ such that $\varnothing \neq \cap_{t=1}^{T} \mathscr{C}_s$ and at least one $x_t \notin \mathscr{C}_t$. This follows due to the density of the reals. Hence, for any $u \in \cap_{t=1}^{T} \mathscr{C}_s$, $\mathrm{Reg}_T^c(u) = \infty$.

## 6. Discussion and Future Work

**Online FPM with Simultaneous Coordination.** The alternating coordination in Algorithm 1 requires that the players are enumerated and know their identity $i$. Otherwise, it is impossible to determine whether $t \mod n = i - 1$. Furthermore, the progress of a player during iterations where $t \mod n \neq i - 1$ can be negligible. Due to these limitations, we note that an online FPM with *simultaneous* shifts of the sets $\mathcal{S}_t^{(i)}$ for all players $i \in [n]$ might be of practical and theoretical interest.

**Extension of the Theoretical Analysis beyond Benign GNEP.** As our experiments illustrate (see Section B in the Appendix), the online FPM converges to a GNE for GNEP beyond the restrictive class of benign GNEP. We leave it to future work to identify further subclasses of GNEP for which convergence can be guaranteed. Identifying subclasses of GNEP with practical relevance and extending the analysis is an interesting future research direction.

**Better Convergence Guarantees.** Note that our convergence analysis does not reflect the progress made by the players in every step $t : t \mod n = i - 1$. As we illustrated in Section 4.1 via an ad-hoc argument, this progress can lead to better convergence in practice. However, this is not reflected in our analysis. Improving the analysis such that it can capture these steps constitutes an interesting future research direction.

**Simultaneous cooperation and non-cooperation games.** Requiring that the players never violate a constraint in a repeated GNEP compels cooperation between the players. At the same time, all players have a selfish interest in minimizing their losses. This can be interpreted as a simultaneous cooperation and competitive game: The competitive aspect of the game is captured by the requirement to optimize the payoffs given by the loss function $\nu^{(i)}\left(x^{(i)}, x^{(-i)}\right)$ in every round. To account for the cooperation aspect, we define a cooperation game based on the constraint violations. A cooperation game with $[n]$ players is defined via a characteristic function that gives the collective payoff for a coalition between $\mathcal{M}$ players with $\mathcal{M} \subset [n]$. Hence, we define a finite variant of the indicator function $\hat{\chi}_{\mathscr{C}}(\cdot) : \mathcal{X} \to \mathbb{R}$ with $\hat{\chi}_{\mathscr{C}}(x) = 0$ if $x \in \mathscr{C}$ and otherwise $\hat{\chi}_{\mathscr{C}}(x) = C$ where $C > 0$ denotes a (large) constant. We let $\psi : 2^{[n]} \to \mathbb{R}$ with $\psi(\varnothing) = 0$ and

$$\psi(\mathcal{M}) = C - \frac{1}{|\mathcal{M}|} \sum_{j \in \mathcal{M}} \hat{\chi}_{\mathcal{X}^{(j)}(x^{(-j)})}\left(x^{(j)}\right).$$

The players can guarantee that this characteristic function $\psi$ is $C$ by forming the *grand coalition*, that is the coalition consisting of all players $[n]$. Choosing $C$ sufficiently large guarantees that the players have an incentive to form a grand coalition. Note, however, that $\psi$ defines a very simplistic cooperation game. Exploring this connection further might be an interesting future research direction.

**Stronger robustness guarantees and relation to regret:** Our regret guarantees are limited to strongly benign GNEP. Further, they are derived from convergence guarantees. It might be an interesting future research direction to extend these guarantees beyond strongly benign GNEP and to derive convergence guarantees (with respect to GNE) from no-regret guarantees for online learning algorithms with time-varying constraints.

## Acknowledgments

Van Erven was supported by the Netherlands Organization for Scientific Research (NWO) under grant number VI.Vidi.192.095. This research was performed while Sachs was also at the Korteweg-de Vries Institute for Mathematics, University of Amsterdam, supported by the Netherlands Organization for Scientific Research (NWO) under grant number VI.Vidi.192.095. At the time of publication, Sachs is funded by the European Union - Next Generation EU, PRIN 2022, CUP:J53D23007170001.

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

## Appendix A. Relations and Examples for Strongly Benign GNEP: Missing Proofs

### Proposition 10

1. *Consider a GNEP with $\mu$-strongly convex loss functions. Further, suppose $\nu^{(i)}(\,\cdot\,, x^{-i})$ are $L$-smooth and there exists a GNE $u \in \mathfrak{C}$ such that $\nabla_i \nu^{(i)}(u^{(i)}, x^{(-i)}) = 0$ for all $x \in \mathfrak{C}$. Then the benign angular condition is satisfied with $\delta = \mu L$.*

2. *Consider a strongly benign GNEP (cf. Definition 2) and assume that the $\nabla_i \nu^{(i)}(\,\cdot\,, x^{-i})$ are $L$-bi-Lipschitz. Then the game is $(\delta/L)$-strongly monotone.*

### Proof

1. Due to $\mu$-strong convexity and $L$-smoothness, we have that for all players $i \in [n]$

$$
\left\langle \nabla_i \nu^{(i)}(x^{(i)}, x^{(-i)}) - \nabla_i \nu^{(i)}(u^{(i)}, x^{(-i)}), x^{(i)} - u^{(i)} \right\rangle \geqslant \mu \left\| x^{(i)} - u^{(i)} \right\|^2
$$
$$
\geqslant \mu L \left\| x^{(i)} - u^{(i)} \right\| \left\| \nabla_i \nu^{(i)}(x^{(i)}, x^{(-i)}) - \nabla_i \nu^{(i)}(u^{(i)}, x^{(-i)}) \right\|.
$$

Using that $\nabla_i \nu^{(i)}(u^{(i)}, x^{(-i)}) = 0$ by assumption and reordering the terms gives the benign angular condition with parameter $\mu L$.

2. Due to the benign angular condition, we have

$$
\left\langle \nabla_i \nu^{(i)}(x^{(i)}, x^{(-i)}), x^{(i)} - u^{(i)} \right\rangle \geqslant \delta \left\| \nabla_i \nu^{(i)}(x^{(i)}, x^{(-i)}) \right\| \left\| x^{(i)} - u^{(i)} \right\|.
$$

We note that for strongly benign GNEP there exists a unique GNE $u \in \mathfrak{C}$ such that for all $x \in \mathfrak{C}$, we have $\nabla_i \nu^{(i)}(u^{(i)}, x^{(-i)}) = 0$. Thus, $\|\nabla_i \nu^{(i)}(x^{(i)}, x^{(-i)})\| = \|\nabla_i \nu^{(i)}(x^{(i)}, x^{(-i)}) - \nabla_i \nu^{(i)}(u^{(i)}, x^{(-i)})\|$ and due to $L$-bi-Lipschitzness $\|\nabla_i \nu^{(i)}(x^{(i)}, x^{(-i)}) - \nabla_i \nu^{(i)}(u^{(i)}, x^{(-i)})\| \geqslant L^{-1}\|x^{(i)} - u^{(i)}\|$. Overall, we obtain

$$
\left\langle \nabla_i \nu^{(i)}(x^{(i)}, x^{(-i)}) - \nabla_i \nu^{(i)}(u^{(i)}, x^{(-i)}), x^{(i)} - u^{(i)} \right\rangle \geqslant \frac{\delta}{L} \left\| x^{(i)} - u^{(i)} \right\|^2.
$$

Summing over all players implies $\frac{\delta}{L}$-strong monotonicity.

∎

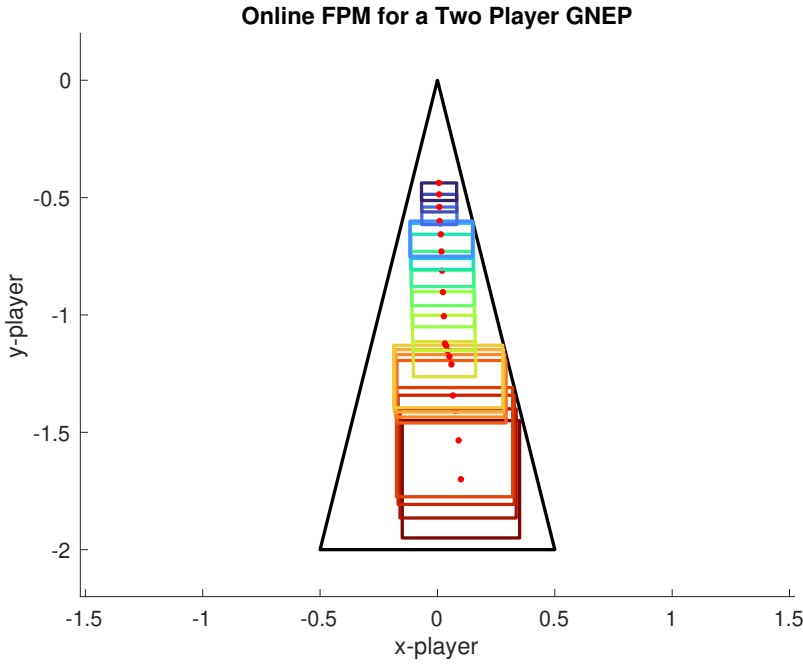

Figure 2: The first 40 iterations of the online FPM Algorithm 1 on a GNEP that does not satisfy Condition 1 in Definition 2. Details in Example 5.

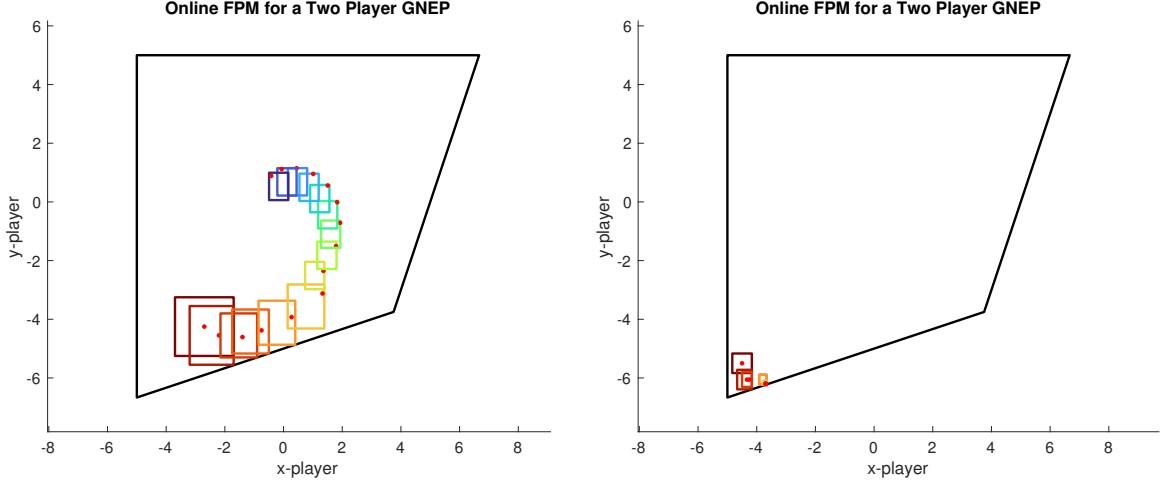

Figure 3: The first 30 iterations of the online FPM on the GNEP defined in Example 6 for two different initializations.

## Appendix B. Additional Examples and Experiments beyond Benign GNEP

In this section, we collect further experiments that show that the online FPM converges to GNE beyond its theoretical guarantees. All GNEP in this section are two-player games with $d^{(1)} = d^{(2)} = 1$. As before, we denote the players' actions by $x, y \in \mathbb{R}$ to eliminate the superscripts. We only display the iterates for every second step in the plots for the benefit of clarity.

We start with an example that satisfies the benign angular condition and has a unique GNE, but the GNE lies on the boundary. Hence, Condition 1 of Definition 2 is violated.

**Example 5 (Non-benign GNEP)** *Consider*

$$\min_{x \in \mathbb{R}} \max_{y \in \mathbb{R}} x^2 - y^2 \qquad s.t. \quad [x, y] \in \mathfrak{C},$$

*with*

$$\mathfrak{C} := \left\{ [x, y] \in \mathbb{R}^2 : y \geqslant -2, \ 4x + y \leqslant 0, \ -4x + y \leqslant 0 \right\}.$$

*The unique GNE is attained at $[\bar{x}, \bar{y}] = [0, 0]$. Note that $[\bar{x}, \bar{y}] \in \mathrm{bd}\,\mathfrak{C}$, hence Condition 1 in Definition 2 is not satisfied. However, as can be seen in Figure 2, the online FPM converges to the unique GNE on the boundary.*

Next, we have an example that does not satisfy Conditions 2 and 3 in Definition 2: that is, there does not exist a $\phi$ such that $D_{\min}(\phi) > 0$ and the angular condition is violated. However, Condition 1 in Definition 2 is satisfied.

**Example 6 (Non-benign GNEP)** *Consider the following GNEP*

$$\min_{x \in \mathbb{R}} \max_{y \in \mathbb{R}} x^2 - 10xy - 2y^2 \qquad s.t. \quad [x, y] \in \mathfrak{C}$$

*with*

$$\mathfrak{C} := \left\{ [x, y] \in \mathbb{R}^2 : x \geqslant -5, \ y \leqslant 5, \ x - \frac{1}{3}y \leqslant 5, \ y - \frac{1}{3}x \geqslant -5 \right\}.$$

*This problem has a GNE at $[0, 0]$. As can be seen in Figure 3, it might converge to the GNE for some initial points. However, initializations exist for which the online FPM converges to a non-equilibrium point as seen in the second plot.*

## Appendix C. Feasibility of the Online FPM: Missing Proofs

We first show the following propositions.

**Proposition 11** [Coordination of Players] *Suppose all players follow Algorithm 1. Then, for all players $i \in [n]$ and all iterations $t \in [T]$, all players are in the same phase $\mathscr{P}_k$.*

**Proof** This follows directly from the observation that the termination criterion **TC** is the same for all players. Hence, it will be either satisfied for all players or for none. ∎

**Theorem 7** [Feasibility] *Suppose all players are following Algorithm 1. Then for all iterations $t \in [T]$, we have $\mathcal{S}_t^{[\times[n]]} \subseteq \mathfrak{C}$; this implies in particular that $[x_t^{(1)}, \ldots, x_t^{(n)}] \in \mathfrak{C}$.*

**Proof** Note that due to the definition of $\iota_t^{(i)}$ and **Vupdate**, $\mathcal{S}_t^{[\times[n]]} \subseteq \mathfrak{C}$ holds. To show the second result, we show that $x_t^{(i)} \in \operatorname{relint} \mathcal{S}_t^{(i)}$ for all $i \in [n]$ and all $t \in [T]$. This follows via induction. More specifically, consider the two cases of the **Update**:

1. Iterations with $(t \bmod n) + 1 \neq i$. Note that in this case the $i^{th}$ player has not updated its set, hence $\mathcal{S}_t^{(i)} = \mathcal{S}_{t-1}^{(i)}$. Since $x_{t-1}^{(i)} \in \mathcal{S}_{t-1}^{(i)}$, the definition of $\bar{\eta}_t^{(i)}$ guarantees that $x_t^{(i)} \in \operatorname{relint} \mathcal{S}_t^{(i)}$.

2. Iterations with $(t \bmod n) + 1 = i$. In this case, the $i^{th}$ player has translated its set, thus $\mathcal{S}_t^{(i)} \neq \mathcal{S}_{t-1}^{(i)}$. Note that due to the definition of **Vupdate** and **Update**, the iterate $x_t^{(i)}$ is translated by the same vector as the set $\mathcal{S}_t^{(i)}$, i.e., $x_t^{(i)} = x_{t-1}^{(i)} + v_t^{(i)}$ and $\mathcal{S}_t^{(i)} = \mathcal{S}_{t-1}^{(i)} + \{v_t^{(i)}\}$. Since $x_{t-1}^{(i)} \in \mathcal{S}_{t-1}^{(i)}$, the claim follows.

$\blacksquare$

## Appendix D. Convergence of the Online FPM: Missing Proofs

### D.1. Technical Results

We first introduce several technical lemmata. We note that most of these results are small variations of existing results. Hence, the following lemmata are primarily included for the convenience of the reader.

**Definition 12 (Moreau envelope)** *Given a function $f : \mathbb{R}^d \to \mathbb{R}$, its Moreau envelope is the function $f^\gamma : \mathbb{R}^d \to \mathbb{R}$ defined by $f^\gamma(x) = \inf_{y \in \mathbb{R}^d} f(y) + \frac{1}{2\gamma} \|y - x\|^2$.*

Some basic properties of the Moreau envelope.

1. $f^\gamma$ is $(1/\gamma)$-smooth.

2. If $\operatorname{argmin}_{x \in \mathcal{X}} f(x)$ exists, then $\operatorname{argmin}_{x \in \mathcal{X}} f^\gamma(x) = \operatorname{argmin}_{x \in \mathcal{X}} f(x)$.

See, Bauschke and Combettes (2011, Proposition 12.29, and 12.30). We use the Moreau envelope for the norm distance function. For completeness, we add the following derivations:

**Proposition 13** *Fix $u \in \mathbb{R}^d$ and $a > 0$, and consider the function $f : \mathbb{R}^d \to \mathbb{R}$ defined by $f(x) = a \|x - u\|$. Then, for any $\gamma > 0$,*

$$f^\gamma(x) = \begin{cases} a \|x - u\| - \frac{\gamma}{2}a^2 & \text{if } a\gamma \leqslant \|x - u\| \\ \frac{1}{2\gamma} \|x - u\|^2 & \text{if } a\gamma > \|x - u\| \end{cases},$$

*and $-\frac{\gamma}{2}a^2 \leqslant f^\gamma(x) - f(x) \leqslant 0$ for all $x \in \mathbb{R}^d$.*

**Proof** By definition of the Moreau envelope,

$$f^\gamma(x) = \min_{y \in \mathbb{R}^d} \left\{ a\,\|y - u\| + \frac{1}{2\gamma}\,\|y - x\|^2 \right\}.$$

We will use that the minimizing $y$ always lies on the line segment $\{(1 - \lambda)x + \lambda u : \lambda \in [0, 1]\}$ between $x$ and $u$. To see this, consider any $y$ not on this line segment. Then we can find a $y'$ that does lie on the line segment with smaller value, by minimizing the second term $\|y' - x\|^2$ subject to not increasing the first term: $\|y' - u\| \leqslant \|y - u\|$. This $y'$ will be the projection of $x$ onto a ball around $u$ of radius $\|y - u\|$, and will hence lie on the line segment between $x$ and $u$. It follows that

$$f^\gamma(x) = \min_{\lambda \in [0,1]} \left\{ a\,\big\|((1 - \lambda)x + \lambda u) - u\big\| + \frac{1}{2\gamma}\,\big\|((1 - \lambda)x + \lambda u) - x\big\|^2 \right\}$$

$$= a\,\|x - u\| + \min_{\lambda \in [0,1]} \left\{ \frac{\lambda^2}{2\gamma}\,\|x - u\|^2 - a\lambda\,\|x - u\| \right\}.$$

The minimizer in $\lambda$ of this quadratic is

$$\lambda^\star = \begin{cases} \frac{a\gamma}{\|x - u\|} & \text{if } a\gamma \leqslant \|x - u\|, \\ 1 & \text{otherwise.} \end{cases}$$

Plugging these expressions in, the first part of the proposition follows. For the second part, we observe that the claim is satisfied for any $x$ such that $a\gamma \leqslant \|x - u\|$. If $\|x - u\| < a\gamma$, then $f^\gamma(x) - f(x) + \frac{\gamma}{2}a^2 = \frac{1}{2\gamma}\|x - u\|^2 - a\|x - u\| + \frac{\gamma}{2}a^2 = \frac{\gamma}{2}(\|x - u\| - a)^2 \geqslant 0$. In the second case, i.e., whenever $a\gamma > \|x - u\|$, we have $f^\gamma(x) - f(x) = \frac{1}{2\gamma}\|x - u\|^2 - a\|x - u\| \leqslant \frac{a\gamma}{2\gamma}\|x - u\| - a\|x - u\| = -\frac{1}{2}a\|x - u\| \leqslant 0$. ∎

The following lemma is a key technical lemma for our convergence guarantees. We note that the step size requirements are dependent on potentially unknown quantities. However, we show in Corollary 15 and 16, that for all cases of interest, the step size requirements are satisfied and can be tuned based on known parameters.

**Lemma 14 (Inexact Gradient Descent)** *Suppose $\tilde{f}_t : \mathbb{R}^d \to \mathbb{R}$ is $\tilde{\mu}$-strongly convex and differentiable. Assume that there exist $u \in \mathbb{R}^d$ and $\delta \in (0, 1]$ such that for any $t \in [T]$*

$$\left\langle \nabla \tilde{f}_t(x_t), x_t - u \right\rangle \geqslant \delta \left\| \nabla \tilde{f}_t(x_t) \right\| \|x_t - u\|. \qquad \textbf{(Angular Condition)}$$

*Consider the update $x_{t+1} = x_t - \eta g_t$ such that $g_t = \nabla \tilde{f}_t(x_t)$ and $\eta > 0$ denotes the stepsize. Assume $\|g_t\| \leqslant G$. Consider $f : \mathbb{R}^d \to \mathbb{R}$ with $f(x) = 2\delta^{-1}\|x - u\|$. Let $\hat{\mu}_t \leqslant \frac{\delta\|g_t\|}{\|x_t - u\|}$. Let $C > 0$ be any constant. Then for any $t \in \mathbb{N}$ and $\eta \leqslant \min\left(\frac{C}{G\sqrt{T}}, \frac{\delta\|x_t - u\|}{\|g_t\|}\right)$*

$$f(x_t) - f(u) \leqslant \Big( f(x_0) - f(u) \Big) \prod_{j=1}^{t-1} \left( 1 - \frac{\hat{\mu}_j \eta}{2} \right) + \frac{C}{\delta^2 \sqrt{T}}. \qquad (3)$$

**Proof** Set $\gamma = \frac{C}{2\sqrt{T}}$ and let $f^\gamma$ denote the Moreau envelope with parameter $\gamma$ of $f$ (See Proposition 13.). Note that due to the basic properties of the Moreau envelope $f^\gamma$ is $(1/\gamma)$-smooth and $f^\gamma(u) = f(u) = 0$. Further, by Proposition 13,

$$f(x_t) \leqslant f^\gamma(x_t) + \frac{\gamma}{2}\frac{4}{\delta^2} = f^\gamma(x_t) + \frac{C}{\delta^2\sqrt{T}}.$$

Because $f^\gamma(x_0) \leqslant f(x_0)$ (again by Proposition 13) and $f(u) = 0$ it is sufficient to show that

$$f^\gamma(x_{t+1}) \leqslant f^\gamma(x_t)\Big(1 - \frac{\hat{\mu}_t\eta}{2}\Big). \tag{4}$$

The result follows by applying (4) recursively. Towards proving (4), we first use $(1/\gamma)$-smoothness of $f^\gamma$ to obtain:

$$f^\gamma(x_{t+1}) \leqslant f^\gamma(x_t) + \langle \nabla f^\gamma(x_t), x_{t+1} - x_t \rangle + \frac{1}{2\gamma}\|x_{t+1} - x_t\|^2$$

$$= f^\gamma(x_t) - \eta\langle \nabla f^\gamma(x_t), g_t \rangle + \frac{\eta^2}{2\gamma}\|g_t\|^2$$

$$= \Big(1 - \frac{\eta\hat{\mu}_t}{2}\Big)f^\gamma(x_t) \underbrace{-\eta\langle \nabla f^\gamma(x_t), g_t \rangle + \frac{\eta^2}{2\gamma}\|g_t\|^2 + \frac{\eta\hat{\mu}_t}{2}f^\gamma(x_t)}_{:=B}.$$

Next, we show that $B \leqslant 0$. We distinguish two cases.

**Case $\|x_t - u\| \geqslant 2\frac{\gamma}{\delta}$.** In this case, we have $f^\gamma(x_t) = 2\delta^{-1}\|x_t - u\| - \frac{2\gamma}{\delta^2} \leqslant 2\delta^{-1}\|x_t - u\|$ by Proposition 13. It follows that $B \leqslant 0$ is guaranteed when

$$-\eta\langle \nabla f^\gamma(x_t), g_t \rangle + \frac{\eta^2}{2\gamma}\|g_t\|^2 + \frac{\eta\hat{\mu}_t}{2}2\delta^{-1}\|x_t - u\| \leqslant 0$$

$$\Leftrightarrow \quad \frac{\eta}{2\gamma}\|g_t\|^2 \leqslant \langle \nabla f^\gamma(x_t), g_t \rangle - \hat{\mu}_t\delta^{-1}\|x_t - u\|.$$

We note that $\nabla f^\gamma(x_t) = \nabla f(x_t) = \frac{2(x_t - u)}{\delta\|x_t - u\|}$. Hence,

$$\langle \nabla f^\gamma(x_t), g_t \rangle = \langle \nabla f(x_t), g_t \rangle = \frac{2}{\delta\|x_t - u\|}\langle x_t - u, g_t \rangle \overset{(1)}{\geqslant} \frac{2}{\delta\|x_t - u\|}\delta\|g_t\|\|x_t - u\| = 2\|g_t\|,$$

where we have used the (**Angular Condition**) for inequality (1) (Recall $g_t = \tilde{\nabla}f_t(x_t)$.). Therefore, to show that $B \leqslant 0$, it is suffices to show that

$$\frac{\eta}{2\gamma}\|g_t\|^2 \leqslant \|g_t\| + \underbrace{\|g_t\| - \hat{\mu}_t\delta^{-1}\|x_t - u\|}_{:=A}$$

The term $A$ is positive for $\hat{\mu}_t$ small enough:

$$\|g_t\| - \hat{\mu}_t\delta^{-1}\|x_t - u\| \geqslant 0 \Leftrightarrow \frac{\|g_t\|\delta}{\|x_t - u\|} \geqslant \hat{\mu}_t,$$

which is satisfied due to the assumption that $\hat{\mu}_t = \frac{\delta\|g_t\|}{\|x_t - u\|}$. It remains to show that

$$\frac{\eta}{2\gamma}\|g_t\|^2 \leqslant \|g_t\|$$

which is satisfied since by assumption $\eta \leqslant \frac{2\gamma}{G}$. Thus, we conclude that $B \leqslant 0$.

**Case $\|x_t - u\| < 2\frac{\gamma}{\delta}$.** Then $f^\gamma(x_t) = \frac{1}{2\gamma}\|x_t - u\|^2$ (c.f. Proposition 13). Hence, $B \leqslant 0$ is guaranteed when

$$-\eta \langle \nabla f^\gamma(x_t), g_t\rangle + \frac{\eta^2}{2\gamma}\|g_t\|^2 + \frac{\eta\hat{\mu}_t}{2}\frac{1}{2\gamma}\|x_t - u\|^2 \leqslant 0$$

$$\Leftrightarrow \quad \frac{\eta}{2\gamma}\|g_t\|^2 \leqslant \langle \nabla f^\gamma(x_t), g_t\rangle - \frac{\eta\hat{\mu}_t}{4\gamma}\|x_t - u\|^2 .$$

Further, using that $\nabla f^\gamma(x_t) = \frac{1}{\gamma}(x_t - u)$ gives

$$\langle \nabla f^\gamma(x_t), g_t\rangle = \frac{1}{\gamma}\langle x_t - u, g_t\rangle \overset{(1)}{\geqslant} \frac{\delta}{\gamma}\|g_t\|\|x_t - u\| ,$$

where inequality (1) is due to the (**Angular Condition**). Therefore, to show that $B \leqslant 0$, it is suffices to show that

$$\frac{\eta}{2\gamma}\|g_t\|^2 \leqslant \frac{\delta}{2\gamma}\|g_t\|\|x_t - u\| + \underbrace{\frac{\delta}{2\gamma}\|g_t\|\|x_t - u\| - \frac{\hat{\mu}}{4\gamma}\|x_t - u\|^2}_{:=A} .$$

Then,

$$A \geqslant 0 \Leftrightarrow \delta\|g_t\| - \frac{\hat{\mu}_t}{2}\|x_t - u\| \geqslant 0 \Leftrightarrow \frac{2\|g_t\|\delta}{\|x_t - u\|} \geqslant \hat{\mu}_t,$$

which is satisfied since $\hat{\mu}_t = \frac{\delta\|g_t\|}{\|x_t - u\|}$. It remains to show that

$$\frac{\eta}{2\gamma}\|g_t\|^2 \leqslant \frac{\delta}{2\gamma}\|g_t\|\|x_t - u\| \Leftrightarrow \eta \leqslant \frac{\delta\|x_t - u\|}{\|g_t\|}, \tag{5}$$

which is satisfied by assumption. This implies $B \leqslant 0$.

Applying (4) recursively gives the claim. ∎

The step size in Lemma 14 seems unusual. However, it reduces to the minimum between $\frac{C}{G\sqrt{T}}$ and $\frac{\delta}{L}$ if the $\tilde{f}_t$'s are $L$-smooth and $u$ is their common minimizer. We note that in this case $g_t^u = \nabla \tilde{f}_t(u) = 0$, hence

$$\|g_t\| = \|g_t - g_t^u\| \overset{(1)}{\leqslant} L\|x_t - u\| .$$

Thus,

$$\frac{\delta\|x_t - u\|}{\|g_t\|} \geqslant \frac{\delta\|x_t - u\|}{L\|x_t - u\|} = \frac{\delta}{L}.$$

Note that the step size $\frac{\delta}{L}$ differs by the constant $\delta$ from the optimal step size of $\frac{1}{L}$ for deterministic gradient descent and $\frac{C}{G\sqrt{T}}$ reduces to the standard (constant) step size choice for online gradient descent whenever $C = D$.

Furthermore, we note that for $g_t^u = 0$, setting $\hat{\mu} = \delta\tilde{\mu}$ implies that $\hat{\mu}_t \leqslant \frac{\delta\|g_t\|}{\|x_t - u\|}$ is satisfied. Indeed, due to strong convexity, we have

$$\mu\|x_t - u\|^2 \leqslant \langle g_t - g_t^u, x_t - u\rangle = \langle g_t, x_t - u\rangle \leqslant \|g_t\|\|x_t - u\| .$$

Thus, $\|x_t - u\| \leqslant \|g_t\|/\mu$.

**Corollary 15** *Suppose all assumptions of Lemma 14 are satisfied. Assume that in addition for all $t \in \mathbb{N}$, $\tilde{f}_t$ are L-smooth and $u = \arg\min_{x \in \mathcal{X}} \tilde{f}_t(x)$ for all $t \in \mathbb{N}$. Set $\hat{\mu} = \delta \tilde{\mu}$. Let $C > 0$ denote a constant and set $\eta = \min(\frac{C}{G\sqrt{T}}, \frac{\delta}{L})$. Then*

$$\Big( f(x_t) - f(u) \Big) \leqslant \Big( 1 - q \Big)^{(t-1)} \Big( f(x_0) - f(u) \Big) + \frac{C}{\delta^2 \sqrt{T}},$$

*where $q = \min\left( \frac{\tilde{\mu}\delta C}{2G\sqrt{T}}, \frac{\delta^2 \tilde{\mu}}{L} \right)$.*

The assumption that all $\tilde{f}_t$'s share a common minimizer is quite strong. Hence, we show a similar corollary under a different assumption: Suppose that for all $t \in \mathbb{N}$, $\left\| \nabla \tilde{f}_t(x_t) \right\| \geqslant \Delta \|x_t - u\|$. Then we obtain the following corollary. Note that we obtain an additive $\epsilon$ term in the bound and the step size and convergence rate are dependent on $\epsilon$.

**Corollary 16** *Suppose all assumptions of Lemma 14 are satisfied. Let $u \in \mathbb{R}^d$ denote the vector for which the **Angular Condition** of Lemma 14 is satisfied. Assume that in addition for all $t \in \mathbb{N}$, $\tilde{f}_t$ are L-smooth and that there exists a $\Delta > 0$ such that $\left\| \nabla \tilde{f}_t(x_t) \right\| \geqslant \Delta \|x_t - u\|$. Set $\hat{\mu} = \Delta \delta$. Let $C > 0$ denote a constant and for any $\epsilon > 0$ set $\eta_t = \min(\frac{C}{G\sqrt{T}}, \frac{\delta\epsilon}{G})$. Then*

$$\Big( f(x_t) - f(u) \Big) \leqslant \Big( 1 - q \Big)^{(t-1)} \Big( f(x_0) - f(u) \Big) + \frac{C}{\delta^2 \sqrt{t}} + 2\delta^{-1}\epsilon,$$

*where $q = \min\left( \frac{\Delta\delta C}{2G\sqrt{T}}, \frac{\delta^2 \Delta\epsilon}{G} \right)$.*

**Proof** First, note that $\hat{\mu}_t = \delta\Delta \leqslant \frac{\delta\Delta\|x_t-u\|}{\|x_t-u\|} \leqslant \frac{\|g_t\|\delta}{\|x_t-u\|}$ due to the assumption $\|\nabla \tilde{f}_t(x_t)\| \geqslant \Delta\|x_t - u\|$. Hence, $\hat{\mu}_t \leqslant \Delta\delta$ implies that the assumptions on $\hat{\mu}_t$ of Lemma 14 are satisfied. Furthermore, we note that for any $x_t$ which is not an $\epsilon$-solution, $\|x_t - u\| \geqslant \epsilon$. Thus, $\frac{\delta\|x_t-u\|}{\|g_t\|} \geqslant \frac{\delta\epsilon}{G} \geqslant \eta$ which implies that the step size assumption of Lemma 14 is satisfied. The claim follows by noting that for any $\epsilon$-solution the bound holds. ∎

**Remark 17 (Related Results)** *Although the convergence result for inexact gradient descent is a technical result tailored to our specific application of benign GNEP, it is interesting to compare it to existing results. Specifically, Khanh et al. (2024) showed convergence for inexact gradient descent under a related condition (cf. (3.1)), capturing the alignment of the inexact gradient and the real gradient with respect to norm differences. While they allow for a more general, non-smooth setting, their gradient approximation needs to converge to the real gradient as $t \to \infty$. Conversely, our analysis explicitly integrates the discrepancy between the real and the inexact gradient in the convergence result, allowing for a problem-dependent convergence result tailored to the application of benign GNEP.*

## D.2. Convergence Results

We apply the technical lemmata from the previous subsection to show the convergence of the iterates to a GNE.

**Theorem 18 (Extended version of Theorem 8)** *Suppose Assumption 2.1 is satisfied and assume all players are following Algorithm 1 with step size $\eta = \min(\frac{D}{G\sqrt{T}}, \frac{\delta}{L})$. Assume we have a $(\phi, \delta)$-strongly benign GNEP. Set*

$$t_0 = \max\left(\frac{4D}{\phi} + 1, \left(\frac{D}{2D_{\min}}\right)^2, \left(\frac{DL}{2G\delta}\right)^2\right).$$

*Then for all players $i \in [n]$ and any $t \in [t_0, T]$*

$$\left\|x_t^{(i)} - u^{(i)}\right\| \leqslant \Xi \left\|x_1^{(i)} - u^{(i)}\right\| \left(1 - \frac{\mu\delta D}{4G\sqrt{T}}\right)^{\frac{t+1}{n}} + \frac{2D}{\delta\sqrt{T}},$$

*where*

$$\Xi = \left(1 - \frac{\mu\delta D}{4G\sqrt{T}}\right)^{-\frac{t_0}{n}}$$

*Suppose we have a $(\delta, \phi, \Delta)$-benign GNEP. Let $\epsilon > 0$ and set*

$$t_0 = \max\left(\frac{4D}{\phi} + 1, \left(\frac{D}{2D_{\min}}\right)^2, \left(\frac{D}{2\epsilon\delta}\right)^2\right).$$

*Then for all players $i \in [n]$ and any $t \in [t_0, T]$*

$$\left\|x_t^{(i)} - u^{(i)}\right\| \leqslant \Xi \left\|x_1^{(i)} - u^{(i)}\right\| \left(1 - \frac{\delta\Delta D}{2G\sqrt{T}}\right)^{\frac{t+1}{n}} + \frac{1}{\delta}\left(\frac{2D}{\sqrt{T}} + 2\epsilon\right),$$

*where*

$$\Xi = \left(1 - \frac{\delta\Delta D}{2G\sqrt{T}}\right)^{\frac{-t_0}{n}}$$

**Proof** We show the convergence result in two steps:

1. We first show that the conditions of Lemma 14 and Corollary 15 (respectively Corollary 16 for benign GNEP) are satisfied, and apply it to the loss function of one player. That is, we apply Lemma 14 with $\tilde{f}_t(\cdot) = \nu(\cdot, x_t^{(i)})$. This gives a bound of the form

$$2\delta^{-1}\left\|x_t^{(i)} - u^{(i)}\right\| \leqslant 2\delta^{-1}\left\|x_1^{(i)} - u^{(i)}\right\| \prod_{s=1}^{t}(1 - \epsilon_s) + \frac{D}{\delta^2\sqrt{T}}.$$

2. In the second step, for any $t \geqslant t_0$, we show that $\epsilon_t$ is sufficiently large to guarantee convergence.

We start by showing that all assumptions of Lemma 14 are satisfied, and then show that the additional assumptions of Corollary 15 and 16 are satisfied for strongly benign and benign GNEP respectively. First, note that due to Assumption 2.1, $\nu^{(i)}(\cdot, x^{-i})$ are $\mu$-strongly convex. Hence, for $\tilde{f}_t(\cdot) = \nu(\cdot, x_t^{(i)})$, the strong convexity assumption of Lemma 14 is satisfied.[5] Next, recall that for

---

5. For the benefit of consistency with respect to the notation used in Assumption 2.1, we keep the notation for the strong convexity parameter of the loss functions $\nu(\cdot, x_t^{(i)})$ as $\mu$ (not $\tilde{\mu}$). When applying Lemma 14, it corresponds to the strong convexity parameter of the function $\tilde{f}$.

benign and strongly benign GNEP the equilibrium is assumed to be unique. Thus, let $(u^{(1)}, \ldots u^{(n)})$ denote this unique GNE. For this unique GNE, by definition, the benign angular condition is satisfied (c.f. Definition 2). That is, there exists a $\delta > 0$ such that

$$\frac{\left\langle \nabla \nu^{(i)}\left(x^{(i)}, x^{(-i)}\right), x^{(i)} - u^{(i)}\right\rangle}{\left\|\nabla \nu^{(i)}\left(x^{(i)}, x^{(-i)}\right)\right\| \left\|x^{(i)} - u^{(i)}\right\|} \geqslant \delta.$$

Thus, the **Angular Condition** of Lemma 14 is satisfied. Next, recall that $\eta = \min(\frac{D}{G\sqrt{T}}, \frac{\delta}{L})$ and by the definition of the **Update** step, a player either takes a gradient descent step with step size $\min(\eta, \bar{\eta}_t^{(i)}/2)$ or $\min(\eta, \iota_t^{(i)})$. In both cases the step size is bounded by $\min(\frac{D}{G\sqrt{T}}, \frac{\delta}{L})$. Setting $\gamma = \frac{D}{2\sqrt{T}}$, this implies that for strongly benign GNEP, the step size conditions of Corollary 15 are satisfied and consequently also the step size requirements of Lemma 14. Further, for a strongly benign GNEP, $\left\|\nabla \nu^{(i)}\left(u^{(i)}, x^{(-i)}\right)\right\| = 0$. Thus, the assumptions for Corollary 15 are satisfied. Setting $\hat{\mu}_t = \frac{\mu\delta}{2}$ for all $t \in \mathbb{N}$ gives for any $k \in \mathbb{N}$ and $t \in \mathscr{P}_k$

$$2\delta^{-1} \left\|x_t^{(i)} - u^{(i)}\right\| \leqslant 2\delta^{-1} \left\|x_1^{(i)} - u^{(i)}\right\| \prod_{s=1}^{t} \left(1 - \frac{\mu\delta\epsilon_s^{(i)}}{4}\right) + \frac{D}{\delta^2\sqrt{T}}, \tag{6}$$

where $\epsilon_s^{(i)} = \min(\eta, \bar{\eta}_t^{(i)}/2)$ if $s \bmod n \neq i - 1$ and $\epsilon_s^{(i)} = \min(\eta, \iota_t^{(i)})$ otherwise. This establishes the first step of the proof.

Next, we show that some of the $\epsilon_t^{(i)}$'s are sufficiently large, i.e., in the order of $1/\sqrt{T}$. We emphasize that this cannot be shown for every iteration, but only for the iteration where $t \bmod n = i - 1$ and $t \geqslant t_0$. Recall that these are the rounds where the $i^{\text{th}}$ player shifts its set $\mathcal{S}_t^{(i)}$. We proceed by inferring that for these iterations, the player's step size can be guaranteed to be in the order of $1/\sqrt{T}$.

Consider any $t \geqslant t_0$, $i \in [n]$ and $x_t^{(i)} \in \mathcal{S}_t^{(i)}$. We denote $g_t^{(i)} = \nabla \nu^{(i)}\left(x_t^{(i)}, x_t^{(-i)}\right)$. Then, due to the definition of $D_{\min}(\phi)$

$$\hat{x}_{t+1}^{(i)} := x_t^{(i)} - \frac{D_{\min}(\phi)}{\left\|g_t^{(i)}\right\|} g_t^{(i)} \in \mathcal{X}^{(i)}(x^{(-i)}).$$

Since the set $\mathcal{X}^{(i)}(x^{(-i)})$ is convex, and since $x_t^{(i)} \in \mathcal{X}^{(i)}(x^{(-i)})$, we know that for any stepsize $\eta \leqslant D_{\min}(\phi)/\left\|g_t^{(i)}\right\|$, $x_t^{(i)} - \eta g_t^{(i)} \in \mathcal{X}^{(i)}(x^{(-i)})$. Due to $t_0 \geqslant (D/(2D_{\min}(\phi)))^2$, we know that for any $t \geqslant t_0$, $\eta = \min(\frac{\delta}{L}, \frac{D}{G\sqrt{T}}) \leqslant \frac{D}{G\sqrt{t}} \leqslant \frac{D_{\min}(\phi)}{2G}$. Hence, $x_{t+1}^{(i)} = x_t^{(i)} - \eta g_t^{(i)} \in \mathcal{X}^{(i)}(x^{(-i)})$. It remains to show that $\mathcal{S}_t^{(i)} - \{\eta g_t^{(i)}\} \subseteq \mathcal{X}^{(i)}(x^{(-i)})$. For this, note that

$$t_0 \geqslant \frac{4D}{\phi} + 1 \geqslant \sum_{j=1}^{\lceil \log_2 \frac{D}{\phi} \rceil} 2^j.$$

Set $k_0 = \lceil \log_2(D/\phi) \rceil$ and recall that due to **TC**, the diameters of the sets $\mathcal{S}_t^{(i)}$ are decreased by $\frac{1}{2}$ at least every $2^k$ iterations. Thus, for any $t \geqslant t_0$,

$$\text{Diameter}\left(\mathcal{S}_t^{(i)}\right) \leqslant \frac{D}{2^{k_0}} \leqslant \phi,$$

and due to the definition of $D_{\min}(\phi)$, $\mathcal{S}_t^{(i)} - \{\eta g_t^{(i)}\} \subseteq \mathcal{X}^{(i)}(x^{(-i)})$. This gives that for any $t \geqslant t_0$

$$2\delta^{-1} \left\| x_t^{(i)} - u^{(i)} \right\| \leqslant 2\delta^{-1} \left\| x_1^{(i)} - u^{(i)} \right\| \prod_{s=t_0}^{t} \left( 1 - \frac{\hat{\mu}_s \eta}{4} \right) + \frac{D}{\delta^2 \sqrt{T}}$$

$$\overset{(1)}{\leqslant} 2\delta^{-1} \left\| x_1^{(i)} - u^{(i)} \right\| \prod_{\substack{s=t_0 \\ (s \bmod n)+1=i}}^{t} \left( 1 - \frac{\mu \delta D}{4G\sqrt{T}} \right) + \frac{D}{\delta^2 \sqrt{T}}$$

$$\leqslant 2\delta^{-1} \left\| x_1^{(i)} - u^{(i)} \right\| \left( 1 - \frac{\mu \delta D}{4G\sqrt{T}} \right)^{\frac{t+1-t_0}{n}} + \frac{D}{\delta^2 \sqrt{T}}$$

$$= \Xi 2\delta^{-1} \left\| x_1^{(i)} - u^{(i)} \right\| \left( 1 - \frac{\mu \delta D}{4G\sqrt{T}} \right)^{\frac{t+1}{n}} + \frac{D}{\delta^2 \sqrt{T}}.$$

Where inequality $(1)$ follows since for any $t \geqslant t_0$ the minimum for the step size definition $\min \left( \frac{D}{G\sqrt{T}}, \frac{\delta}{L} \right)$ is attained at $\frac{D}{G\sqrt{T}}$. Further, we used here that $\hat{\mu}_s = \frac{\mu \delta}{2}$.

For the second part of the result, we note that the conditions of Corollary 16 are satisfied. Indeed, the only difference is the assumption that gradient norms satisfy $\left\| \nabla \nu^{(i)}\left( u^{(i)}, x^{(-i)} \right) \right\| \geqslant \Delta \left\| x_t^{(i)} - u^{(i)} \right\|$, which is satisfied due to the assumptions for benign GNEP. Set $\hat{\mu} = \delta \Delta$. Then analogously to (6),

$$2\delta^{-1} \left\| x_t^{(i)} - u^{(i)} \right\| \leqslant 2\delta^{-1} \left\| x_1^{(i)} - u^{(i)} \right\| \prod_{s=1}^{t} \left( 1 - \frac{\delta \Delta \eta}{2} \right) + \frac{4GD}{\delta^2 \sqrt{T}} + 2\frac{\epsilon}{\delta}. \tag{7}$$

Noting that the rest of the argument is analogous and differs only with respect to the constants, gives the second result. ∎

# Appendix E. Regret Bounds

**Theorem 9** *Suppose Assumption 2.1 is satisfied and assume we have a $(\phi, \delta)$-strongly benign GNEP. Let $\Xi$ and $t_0$ be defined as in Theorem 8. If all players are following Algorithm 1, then for all players $i \in [n]$*

$$\sum_{t=1}^{T} \chi_{\mathcal{X}^{(i)}(x_t^{(-i)})} \left( x_t^{(i)} \right) = 0,$$

*and for all $T \in \mathbb{N}$*

$$\sum_{t=1}^{T} \nu^{(i)} \left( x_t^{(i)}, x_t^{(-i)} \right) - \sum_{t=1}^{T} \nu^{(i)} \left( u^{(i)}, x_t^{(-i)} \right) \leqslant DG \left( \sqrt{T} \left( 2\Xi n \frac{G}{\mu D} + \frac{2}{\delta} \right) + t_0 \right).$$

**Proof** Due to convexity and since by assumption $\|\nabla\nu^{(i)}(x_t^{(i)}, x_t^{(-i)})\| \leqslant G$, we have

$$
\nu^{(i)}\left(x_t^{(i)}, x_t^{(-i)}\right) - \nu^{(i)}\left(u^{(i)}, x_t^{(-i)}\right) \leqslant \left\langle \nabla\nu^{(i)}\left(x_t^{(i)}, x_t^{(-i)}\right), x_t^{(i)} - u^{(i)}\right\rangle
$$
$$
\leqslant \left\|\nabla\nu^{(i)}\left(x_t^{(i)}, x_t^{(-i)}\right)\right\| \left\|x_t^{(i)} - u^{(i)}\right\|
$$
$$
\leqslant G\left\|x_t^{(i)} - u^{(i)}\right\|.
$$

Summing over $t_0$ to $T$ rounds, we obtain

$$
\sum_{t=t_0}^{T} \nu^{(i)}\left(x_t^{(i)}, x_t^{(-i)}\right) - \nu^{(i)}\left(u^{(i)}, x_t^{(-i)}\right) \leqslant \frac{\delta G}{2}\sum_{t=t_0}^{T} 2\delta^{-1}\left\|x_t^{(i)} - u^{(i)}\right\|.
$$

Next, we apply Theorem 8 to all terms with $t \geqslant t_0$. This gives

$$
\sum_{t=t_0}^{T} \nu^{(i)}\left(x_t^{(i)}, x_t^{(-i)}\right) - \nu^{(i)}\left(u^{(i)}, x_t^{(-i)}\right) \leqslant \frac{\delta G}{2}\left(D\Xi\sum_{t=t_0+1}^{T}\left(1 - \frac{\delta\mu D}{4G\sqrt{T}}\right)^{\frac{t+1}{n}} + \frac{2D\sqrt{T}}{\delta}\right)
$$
$$
\leqslant \frac{\delta G D\Xi}{2}\int_{t_0}^{T}\left(1 - \frac{\delta\mu D}{4G\sqrt{T}}\right)^{\frac{x+1}{n}}dx + DG\sqrt{T}
$$
$$
= \frac{\delta G D\Xi}{2}\left(n\frac{\left(\left(1 - \frac{\delta\mu D}{4G\sqrt{T}}\right)^{\frac{t_0+1}{n}} - \left(1 - \frac{\delta\mu D}{4G\sqrt{T}}\right)^{\frac{T+1}{n}}\right)}{-\log\left(1 - \frac{\delta\mu D}{4G\sqrt{T}}\right)}\right) + DG\sqrt{T}
$$
$$
\overset{(1)}{\leqslant} \frac{\delta G D\Xi}{2}\left(n\frac{4G\sqrt{T}}{\delta\mu D}\right) + DG\sqrt{T}.
$$

Where we applied that for all $x > -1$ it holds that $\log(1+x) \leqslant x$. Furthermore, recall that $t_0$ is a constant independent of $T$. Hence, we bound the first $t_0$ rounds using convexity and the Cauchy-Schwarz inequality:

$$
\sum_{t=1}^{t_0} \nu^{(i)}\left(x_t^{(i)}, x_t^{(-i)}\right) - \nu^{(i)}\left(u^{(i)}, x_t^{(-i)}\right) \leqslant \sum_{t=1}^{t_0}\left\langle \nabla\nu^{(i)}\left(x_t^{(i)}, x_t^{(-i)}\right), x_t^{(i)} - u^{(i)}\right\rangle
$$
$$
\leqslant \sum_{t=1}^{t_0}\left\|\nabla\nu^{(i)}\left(x_t^{(i)}, x_t^{(-i)}\right)\right\|\left\|x_t^{(i)} - u^{(i)}\right\|
$$
$$
\leqslant t_0 GD,
$$

which shows the claim. ∎

### E.1. Sublinear Regret for Constrained Online Convex Optimization

**Lemma 19** *Let $S_t \subset \mathbb{R}^d$, $t = 1, 2, \ldots$ be closed, non-empty, convex sets of diameter at most* $\mathrm{Diameter}(S_t) \leqslant D$ *and such that the maximum distances between consecutive sets are small:*

$$
\max_{a \in S_t}\mathrm{dist}(S_{t+1}, a) \leqslant \omega_{t+1}
$$

*for some $\omega_2, \omega_3, \ldots$ Consider online gradient descent with $x_1 \in S_1$ and $x_{t+1} = \mathrm{Proj}_{S_{t+1}} (x_t - \eta_t g_t)$.*
*If the gradients $g_t = \nabla f_t(x_t)$ are uniformly bounded by $\|g_t\| \leqslant G$, the functions $f_t : \mathbb{R}^d \to \mathbb{R}$ are*
*convex and the step-sizes $\eta_1 \geqslant \cdots \geqslant \eta_T > 0$ are non-increasing. Then, for any $u \in \mathbb{R}^d$,*

$$\mathrm{Reg}_T^f(u) \leqslant \frac{1}{2\eta_T}(D + \max_{t \leqslant T} \mathrm{dist}(S_t, u))^2 + G^2 \sum_{t=1}^{T} \frac{\eta_t}{2} + \sum_{t=1}^{T-1} \left( \frac{\omega_{t+1}}{\eta_t} + G \right) \mathrm{dist}(S_{t+1}, u) - \frac{1}{2\eta_T}\|x_{T+1} - u\|^2.$$

*In particular, if $\mathrm{dist}(S_t, u) \leqslant \frac{c}{\sqrt{t}}$, $\eta_t = \frac{D+c}{G\sqrt{t}}$ and $\omega_{t+1} \leqslant c'G\eta_t$, then*

$$\mathrm{Reg}_T^f(u) \leqslant \frac{3D + (6 + 4c')c}{2} G\sqrt{T}.$$

*If the functions $f_t : \mathbb{R}^d \to \mathbb{R}$ are also $\mu$-strongly convex and the step-sizes $\eta_t = \frac{1}{t\mu}$, then, for any*
*$u \in \mathbb{R}^d$,*

$$\mathrm{Reg}_T^f(u) \leqslant \frac{G^2}{\mu} \log T + \sum_{t=1}^{T-1} \left( \frac{\omega_{t+1}}{\eta_t} + G \right) \mathrm{dist}(S_{t+1}, u) - \frac{1}{2\eta_T}\|x_{T+1} - u\|^2.$$

**Proof** The proof follows the standard OGD analysis, except that we need to be more careful when applying the Pythagorean inequality, because it may be the case that $u \notin S_t$. Let $\tilde{x}_{t+1} = x_t - \eta_t g_t$ be the unprojected update, and let $u_t = \mathrm{Proj}_{S_t}(u)$. Then

$$
\begin{aligned}
\|\tilde{x}_{t+1} - u\|^2 &= \|\tilde{x}_{t+1} - u_{t+1} + u_{t+1} - u\|^2 \\
&= \|\tilde{x}_{t+1} - u_{t+1}\|^2 + \|u_{t+1} - u\|^2 + 2 \langle \tilde{x}_{t+1} - u_{t+1}, u_{t+1} - u \rangle \\
&\geqslant \|\tilde{x}_{t+1} - x_{t+1}\|^2 + \|x_{t+1} - u_{t+1}\|^2 + \|u_{t+1} - u\|^2 + 2 \langle \tilde{x}_{t+1} - u_{t+1}, u_{t+1} - u \rangle \\
&= \|\tilde{x}_{t+1} - x_{t+1}\|^2 + \|x_{t+1} - u\|^2 - 2 \langle x_{t+1} - u_{t+1}, u_{t+1} - u \rangle + 2 \langle \tilde{x}_{t+1} - u_{t+1}, u_{t+1} - u \rangle \\
&= \|\tilde{x}_{t+1} - x_{t+1}\|^2 + \|x_{t+1} - u\|^2 + 2 \langle \tilde{x}_{t+1} - x_{t+1}, u_{t+1} - u \rangle \\
&\geqslant \|\tilde{x}_{t+1} - x_{t+1}\|^2 + \|x_{t+1} - u\|^2 - 2\|\tilde{x}_{t+1} - x_{t+1}\|\|u_{t+1} - u\| \\
&\geqslant \|x_{t+1} - u\|^2 - 2\|\tilde{x}_{t+1} - x_{t+1}\|\|u_{t+1} - u\|,
\end{aligned}
$$

where the first inequality follows by the Pythagorean inequality, and the second one by Cauchy-Schwarz. Observing that

$$
\begin{aligned}
\|\tilde{x}_{t+1} - x_{t+1}\| &\leqslant \|\tilde{x}_{t+1} - \mathrm{Proj}_{S_{t+1}}(x_t)\| = \|x_t - \mathrm{Proj}_{S_{t+1}}(x_t) - \eta_t g_t\| \\
&\leqslant \|x_t - \mathrm{Proj}_{S_{t+1}}(x_t)\| + \eta_t\|g_t\| \leqslant \omega_{t+1} + \eta_t\|g_t\|,
\end{aligned}
$$

we conclude that

$$\|\tilde{x}_{t+1} - u\|^2 \geqslant \|x_{t+1} - u\|^2 - 2(\omega_{t+1} + \eta_t\|g_t\|) \mathrm{dist}(S_{t+1}, u).$$

It follows that

$$\sum_{t=1}^{T} f_t(x_t) - f_t(u) \leqslant \sum_{t=1}^{T} \langle x_t - u, g_t \rangle$$

$$= \sum_{t=1}^{T} \frac{1}{2\eta_t} \left( \|x_t - u\|^2 - \|\tilde{x}_{t+1} - u\|^2 \right) + \sum_{t=1}^{T} \frac{\eta_t}{2} \|g_t\|^2$$

$$\leqslant \sum_{t=1}^{T} \frac{1}{2\eta_t} \left( \|x_t - u\|^2 - \|x_{t+1} - u\|^2 + 2(\omega_{t+1} + \eta_t \|g_t\|) \operatorname{dist}(S_{t+1}, u) \right) + \sum_{t=1}^{T} \frac{\eta_t}{2} \|g_t\|^2$$

$$= \sum_{t=1}^{T} \frac{1}{2\eta_t} \left( \|x_t - u\|^2 - \|x_{t+1} - u\|^2 \right) + \sum_{t=1}^{T} \left( \frac{\omega_{t+1}}{\eta_t} + \|g_t\| \right) \operatorname{dist}(S_{t+1}, u) + \sum_{t=1}^{T} \frac{\eta_t}{2} \|g_t\|^2$$

$$= \frac{1}{2\eta_1} \|x_1 - u\|^2 - \frac{1}{2\eta_T} \|x_{T+1} - u\|^2 + \sum_{t=2}^{T} \left( \frac{1}{2\eta_t} - \frac{1}{2\eta_{t-1}} \right) \|x_t - u\|^2$$

$$+ \sum_{t=1}^{T} \left( \frac{\omega_{t+1}}{\eta_t} + \|g_t\| \right) \operatorname{dist}(S_{t+1}, u) + \sum_{t=1}^{T} \frac{\eta_t}{2} \|g_t\|^2$$

$$\leqslant \frac{(D + \max_{t \leqslant T} \operatorname{dist}(S_t, u))^2}{2\eta_T} - \frac{1}{2\eta_T} \|x_{T+1} - u\|^2$$

$$+ \sum_{t=1}^{T} \left( \frac{\omega_{t+1}}{\eta_t} + G \right) \operatorname{dist}(S_{t+1}, u) + G^2 \sum_{t=1}^{T} \frac{\eta_t}{2}.$$

Since $S_{T+1}$ is not used in the algorithm, we assume that $S_{T+1} = \mathbb{R}^d$ without loss of generality, so that $\operatorname{dist}(S_{T+1}, u) = 0$. The first result of the theorem then follows.

For the second result we plug in the extra assumptions, which gives

$$\operatorname{Reg}_{\mathrm{T}}^{\mathrm{f}}(u) \leqslant \frac{(D + c)G}{2} \sqrt{T} + \frac{(D + (3 + 2c')c)G}{2} \sum_{t=1}^{T} \frac{1}{\sqrt{t}}.$$

Using that $\sum_{t=1}^{T} \frac{1}{\sqrt{t}} \leqslant 2\sqrt{T} - 1 \leqslant 2\sqrt{T}$, the second result then follows.

Next, we show the regret bound for strongly convex functions. As usual, we use that $\frac{1}{2\eta_t} - \frac{\mu}{2} = \frac{1}{2\eta_{t-1}}$. Note that for $t = 1$, this is zero.

$$\sum_{t=1}^{T} f_t(x_t) - f_t(u) \leqslant \sum_{t=1}^{T} \langle x_t - u, g_t \rangle - \frac{\mu}{2} \|x_t - u\|^2$$

$$\leqslant \sum_{t=1}^{T} \left( \left( \frac{1}{2\eta_t} - \frac{\mu}{2} \right) \|x_t - u\|^2 - \frac{1}{2\eta_t} \|x_{t+1} - u\|^2 \right)$$

$$+ \sum_{t=1}^{T} \left( \frac{\omega_{t+1}}{\eta_t} + \|g_t\| \right) \operatorname{dist}(S_{t+1}, u) + \sum_{t=1}^{T} \frac{\eta_t}{2} \|g_t\|^2$$

$$= \sum_{t=2}^{T} \frac{1}{2\eta_{t-1}} \|x_t - u\|^2 - \sum_{t=1}^{T} \frac{1}{2\eta_t} \|x_{t+1} - u\|^2$$

$$+ \sum_{t=1}^{T} \left( \frac{\omega_{t+1}}{\eta_t} + \|g_t\| \right) \operatorname{dist}(S_{t+1}, u) + \sum_{t=1}^{T} \frac{\eta_t}{2} \|g_t\|^2$$

$$= -\frac{1}{2\eta_T} \|x_{T+1} - u\|^2 + \sum_{t=1}^{T} \left( \frac{\omega_{t+1}}{\eta_t} + \|g_t\| \right) \operatorname{dist}(S_{t+1}, u) + \sum_{t=1}^{T} \frac{\eta_t}{2} \|g_t\|^2.$$

Together with the bound $\sum_{t=1}^{T} \frac{\eta_t}{2} \|g_t\|^2 \leqslant \frac{G^2}{\mu} \log T$ and assuming again without loss of generality that $S_{T+1} = \mathbb{R}^d$, the last result follows. ∎

