# OpenReview forum: "An Online Feasible Point Method for Benign Generalized Nash Equilibrium Problems."
_algorithmiclearningtheory.org/ALT/2025/Conference — ALT 2025_

### Official Review · Reviewer_K8UB · 2024-11-05
**An interesting paper but some clarifications are needed**

**Rating:** 6
**Confidence:** 4

**Review:**

This paper considers a repeatedly played generalized Nash equilibrium game. This induces a multi-agent online learning problem with joint constraints. An important challenge in this setting is that the feasible set for each agent depends on the simultaneous moves of the other agents and, therefore, varies over time. As a consequence, the agents face time-varying constraints, which are not adversarial but rather endogenous to the system. Prior work in this setting focused on convergence to a feasible solution in the limit via integrating the constraints in the objective as a penalty function. However, no existing work can guarantee that the constraints are satisfied for all iterations while simultaneously guaranteeing convergence to a generalized Nash equilibrium. This is a problem of fundamental theoretical interest and practical relevance. The authors introduced a new online feasible point method. Under the assumption that limited communication between the agents is allowed, this method guarantees feasibility. They also identify the class of benign generalized Nash equi- librium problems, for which the convergence of our method to the equilibrium is guaranteed. They set this class of benign generalized Nash equilibrium games in context with existing definitions and illustrate our method with examples.

Pros: This paper gives an algorithm for the updates of the desired sets and the iterates. The contributions are summarized as follows: 1. The authors introduce the online feasible point method (online FPM) and show that feasibility is guaranteed for each iteration if each player uses the online FPM in a repeated generalized game. This method follows a fundamentally different approach from existing methods which are based on Lagrangian splitting schemes or penalty methods. 2. The authors identify a subclass of GNEP, which are called strongly benign GNEP, for which convergence of the iterates to an equilibrium is guaranteed. Furthermore, they illustrate the class of strongly benign GNEP with examples and set it into context with common assumptions such as strong monotonicity for games. 3. The authors derive regret bounds for each player and set these results in context with existing results on online learning with varying constraints. For online problems with varying constraints, strong guarantees are possible if the constraints are not adversarial but endogenous to the system. 4. The authors illustrate the behavior of the proposed method with various examples of strongly benign GNEP. They also demonstrate the convergence to the GNEP without constraint violation for GNEP beyond strongly benign GNEP.

Cons: First of all, the authors did not explain why it is important to study generalized Nash equilibrium problems in online setting. I agree that the multi-agent online learning with joint constraints would be a natural extension but some justification from practical sides is necessary. Second, the proposed method is simple and easily understood. However, when the constraint set is complicated, I do not know if we can implement some steps in pratice. For example, it would be not easy to compute (1) the minimal distance with respect to the gradient direction $g_t^{(i)}$ of the set $S_t^{\times [n]}$ to the boundary of feasible action set $\mathcal{C}$ and (2) the maximal step into the gradient direction $g_t^{(i)}$ while staying within $S_t^{(i))}$. In my humble opinion, it is computationally expensive to maintain the feasibility of the iterates for solving generalized Nash equilibrium problems mostly because the constraint sets are complicated. This is why the previous works integrate the constraints in the objective as a penalty function. While I definitely agree that the feasibility would be practically relevant, it would be better if the authors clarify why the proposed methods are implementable for solving practical problems. Finally, your results in Theorem 7 demonstrate the last-iterate convergence rate in the $\ell_2$-norm which only holds under the strongly monotonicity property and its variants. Thus, I am not sure if a class of strongly benign GNEPs are a unnecessary relaxation of strongly monotone GNEPs. I encourage the authors to provide some nontrivial examples which are strongly benign GNEPs but not strongly monotone GNEPs.

**Paper Award:**

No

---

> ### Author Response · Authors · 2024-11-25
>
> We thank the reviewer for the interesting questions and helpful comments. Below, we address all points in detail:
>
> Motivations for Nash-equilibria in an online setting:
>
> Since multi-agent online learning is a
> computationally attractive standard approach for approximating regular Nash
> equilibria, we believe it is natural to study if/when this approach can be
> extended to generalized Nash equilibria problems.
> For practical relevance, we kindly refer the reviewer to our example on page 2, section 'practical relevance'. Note that this example indeed defines a repeatedly played generalized game.
>
>
>  Computational feasibility:
>
> The computational costs depend on two aspects: (1) the complexity of the constraint set $C$ and (2) the complexity of $ S_t^{(i)}$. Note that the $ S_t^{(i)}$'s are part of the algorithm definition, whereas $C$ is part of the problem definition. Thus, with the choices of $ S_t^{(i)}$, we can reduce the computational cost.
>
>    For example, if $ S_t$ is defined as the convex hull of a finite number of $k$ points (e.g. a hypercube), feasibility can be checked by a line search in the gradient direction for a finite number of points. To illustrate this further, assume $C$ is a polytope defined by $m$ affine constraints. Then the
>  line search reduces to checking for the intersections of a finite number of pairs of lines which can be computed in $O(m k \cdot \mathrm{poly}(d) )$ where $d$ denotes the dimension (formal argument below). Thus, in this case, we suffer a mild computational overhead.
>
>  However, if $ S_t$ is not chosen as a polytope, ensuring feasibility can be computationally expensive as the reviewer pointed out correctly.
>
>
>  Lemma: Let $C = \{ x \in R^d: \sum_{i=1}^m \langle a_i,x\rangle \leq c_i \} $ for $a_i \in R^d$ and $c_i \in R$.  Suppose $C \neq \emptyset$ and let $\mathcal{S} = \mathrm{conv}(p_1, \dots p_k) $ be the convex hull of $k$ points with $\mathcal{S} \subset C$. Then we can compute $\gamma \geq 0$ such that  $ S + \{\gamma v\} \subset C$, and this computation has  $ O(mk \cdot \mathrm{poly}(d))$ complexity.
>
>  Proof: We note that there are $m k$ pairs of points $p_i$ and constraints $\langle a_j,x\rangle \leq c_j$. For any $i \in [k] , j\in [m] $ let  $(p_i, (a_j,c_j))$ denote such a pair and denote by $a_j^{\perp}$ the orthogonal vector to $a_j$, i.e., $\langle a_j, a_j^{\perp}\rangle = 0$. The intersection of the two lines $p_i + \alpha_i v$ and $c_j + \beta_j a_j^{\perp}$ with $\alpha_i, \beta_j \in R$ can be computed by solving a linear equation with variables $\alpha_i, \beta_j$. For each pair $(i,j)$ we denote the solution of this linear equation by $(\alpha^*_i,\beta_j^*)$.
> We set $\gamma \geq 0$ to the minimal $\alpha_i^\star$ over all pairs, that is $\gamma = \min( \alpha^*_i : (\alpha^*_i,\beta^*_j)$ for $ i\in [k], j \in [m] )$.  Since  $\mathcal{S}$ is defined as the convex hull of $p_1, \dots p_k$, this ensures the first claim $\mathcal{S} +\gamma \{v \}\subset C$.
> The second claim follows by observing that a linear equation can be solved in $O(\mathrm{poly}(d))$ time. Thus, for all pairs the complexity is $O(mk \cdot \mathrm{poly}(d))$.
>
>
>
>   Nontrivial examples which are strongly benign GNEPs but not strongly monotone GNEPs.:
>
>   There are two aspects to this question: (1) setting the class of strongly benign and strongly monotone GNEP in relation, and (2) the assumptions for the last step convergence results in Theorem 7.
>
> Point (1): Please note that we do not claim that the class of strongly benign GNEP is a weaker
>    assumption than strong monotonicity. Indeed, Proposition~5 shows that,
>    under an additional bi-Lipschitzness assumption, our strongly benign
>    condition implies strong monotonicity. Conversely, specific strongly monotone games (cf. Proposition 5, (1)) are also strongly benign. Hence, the class of strongly monotone games and strongly benign games do have a non-trivial intersection, but neither is a subset of the other.
>    An example of a strongly benign GNEP, which is not strongly monotone, can be constructed based on the results in [4].
>
>    Indeed, they provide an example of a game on a bounded domain in which a Restricted
>    Secant Inequality holds, and in which the gradients are Lipschitz. These conditions
>    together imply the angular condition. On the other hand, the utilities of the players
>    are not convex, so the game is not strongly monotone.
>
>
>    Point (2): As the reviewer pointed out, Theorem 7 only holds under the strong monotonicity assumption (Assumption 2.1 (2), assumed for all our results). The restriction to strongly benign GNEP is needed since strong monotonicity does not imply the angular condition (cf. Prop. 5 (1)) and the angular condition is essential for Lemma 12.
>
>
>
>
>    [4]: Tracking solutions of time-varying variational inequalities. Hedi Hadiji, Sarah Sachs, Cristobal Guzman, 2024.

---

### Official Review · Reviewer_hVaj · 2024-11-08

**Rating:** 6
**Confidence:** 3

**Review:**

This paper studies repeatedly Generalized Nash Equilibrium Problem (GNEP) a game theoretic setting where the constraints of each agent also depends on the actions of alternative agents. For such a game the authors design two algorithms that provably converge to a Nash equilibrium while not violating the constraints throughout the path. Such strong result cannot hold for harder constraint settings, e.g., online learning with adversarial time varying constraints.

Technically introduce  anew classes of models which put additional assumption on the GNEP structure: the Strongly Benign GNEP (definition 2) an Benign GNEP(definition 3). The authors establish the existence of such structure under natural assumptions (e.g., smoothness and strong convexity of the loss functions).

Next, the authors provide the naive algorithm that updates the agent policies in a round-robin way and present their main algorithm, Alg. 1 Alg. 1, intuitively, improves over the naive approach since it allows for simultaneous updates of agent. The authors show Alg. 1 converges with rate of 1/\sqrt{T} to a Nash equilibrium while satisfying the constraints during the application of the algorithm (assuming all agents follow Alg. 1).

Finally, the authors review the connection of their new result to existing art, and, especially, to online learning with time varying constraints -- which is a harder problem, in which satisfying the constraints and achieving \sqrt{T} rate is impossible.

Opinion.
I found the result interesting and novel. The fact Alg. 1 does not violate the dependent constraints while achieving \sqrt{T} convergence is surprising, to the best of my knowledge. Nevertheless, the setting the authors study is somewhat specialized (e.g., the assumptions the authors made, and the way the constraint set is constructed as an average of other agents' action). Additionally, the proofs slightly lacks clarity in my opinion (I did not follow the feasibility proof the authors supplied. See questions below). Would be happy to increase my score if the authors can provide clarifications for my questions below.

Question.
1. Even though the theoretical appeal, is the constraints studied in this work are relevant for practical applications? Namely, when do we expect to have time varying constraints in form of the setting in Section 2.1?
2. The benign angular condition is not properly explained in my opinion. In which cases we expect this condition to hold? In which cases it won't hold? Can the authors supply few examples for intuition? Possibly providing a discussion about it after Proposition 5 can be helpful.
3. In the proof of Theorem 6: how come x_t-1^{(i)}\in S_{t-1} implies the result? We would like to show feasability of all the agents' vectors. I would ask the authors to expend on the current proof as it seems to me incomplete.
4. Are there any variations of lemma 12 in literature? Additionally, is the angular condition was studied by prior works?
5. Lemma 12, if i understand correctly, is weaker than standard results when it comes to smooth and strongly convex functions due to the 1/\sqrt{T} term. Could you elaborate on this? What is the source of such additional term comparing to other existing results?

Comments.
1. The authors use nabla in places it would be more proper to use nabla_{x^{(i)}}, namely, when taking gradients only wrt to x^{(i)} (see, for example, definition of D_min^{(i)}(\phi).

**Paper Award:**

No

---

> ### Author Response · Authors · 2024-11-25
>
> We thank the reviewer for the helpful comments and interesting questions. Below, we address all your points in detail:
>
> (Q1) Application examples:
>
>  Please note that the setting introduced in Section 2.1 is standard and considered in several previous works. Specifically, the work by Facchini and Kanzow 2007 'Generalized nash equilibrium problems' considers the same setting and contains several application examples (cf. Section 2.1-2.3). In particular, their example in Section 2.1 shows that the time-varying constraints that arise from the shared constraints in a GNEP are omnipresent in economic models based on real-world problems.
>    Furthermore, we provide a basic example in Section 1, Paragraph 'Practical example') with a strong motivation for avoiding constraint violations.  We will add a clarifying remark with a reference to the relevant literature.
>
> (Q2) Angular condition:
>
> Please note that we included several examples in Appendix A1. We will migrate them to the main part of the paper and thank the reviewer for pointing out the need for illustrating examples.
>
>   (Q3)  Proof of Thm 6:
>
>  The important point here is that $x_{t}$ and $ S_t$ are translated by the same vector. Thus, if $x_t$ was in $ S_t$, then $x_{t+1}$ will be in the translated set $ S_{t+1}$. Formally, note that $ S_{t+1} =  S_t + \{v\}$ (where $+$ denotes Minkowski addition) and $x_{t+1} = x_t + v$ for the same vector $v$. Hence, if $x_t \in  S_t$ then $x_{t+1} = x_t + v \in  S_t + \{v\} =  S_{t+1}$. We will add this calculation to the proof and thank the reviewer for pointing us to the needed clarification.
>
>   (Q4) Variants of Lemma 12 in the literature/angular condition new?:
>
>   The angular condition appeared naturally to us in the analysis of examples we studied in the context of GNEP,
>  with a geometrically intuitive appeal (Appendix A.1). To our knowledge, such a condition has not been studied before in the context of generalized games.
>  Regarding the inexact gradient descent method, our results (Lemma 12 and corollaries) are very specifically
>  tailored to our needs and we could not find an existing result that would fit our assumptions.
>  However, there indeed exist results in the literature that consider (deterministic) inexact gradient descent (e.g. [1,2,3]), and in particular [2] consider a similar angular condition as we did. However, note that the angular condition considered in this work (3.1) differs from ours, and specifically, these results do not integrate the quantitative angular error in the analysis and instead require a diminishing error as the algorithm proceeds. (Note that $\epsilon_{k} \leq \theta^k \epsilon_1$ in their definition.)  Hence, these results are not applicable to showing convergence in the benign GNEP setting.
>  We will add a discussion after Lemma 12 in the Appendix addressing these related results.
>
>
>
>
>
>  (Q5) Extra $1/\sqrt{T}$ term in Lemma 12:
>
>  Technically the $1/\sqrt{T}$ term comes from the stepsize choice. Our stepsize is intuitively the minimal stepsize between the online and (approximate) offline stepsize (see Corollary 13). The online part of the stepsize choice ($C/(G\sqrt{T})$ inevitably implies a $1/\sqrt{T}$ term in the convergence bound. We note that the stepsize choice is unusual. However, it is motivated by our application, i.e., Lemma 12 is a technical result, and we primarily want to apply it to show the convergence (Thm. 6). Thus, the stepsize in Lemma 12 has to match with the stepsize choices for the algorithm. Further, the stepsize choices of the algorithm are carefully chosen to enable us to show feasibility, control the termination criterion TC, and allow sublinear convergence simultaneously. Trading off all these factors requires the chosen step size.
>
>  Literature:
> [1]: Gradient Convergence in Gradient Methods with Errors,  Dimitri P. Bertsekas and John N. Tsitsiklis, Siam Opt. 2000.
> [2]: A New Inexact Gradient Descent Method with Applications to Nonsmooth Convex Optimization' by Pham Duy Khanh, Boris S. Mordukhovich, Dat Ba Tran, 2024.
> [3]: Universal gradient methods for convex optimization problems, Yu. Nesterov, Math. Program., 2015

---

### Official Review · Reviewer_tiKx · 2024-11-09
**Review Submission 95**

**Rating:** 6
**Confidence:** 3

**Review:**

*Paper Summary*

The paper considers distributed, agent-based algorithms for generalized Nash Equilibrium (GNE). GNE is a natural extension of Nash Equilibrium (NE) in games where agents face coupled constraints, meaning the action set of each agent depends not only on their own actions but also on the actions of others. The paper investigates distributed algorithms that enable agents to reach GNE via limited communication. Specifically, the paper makes the following contributions:

1. The authors propose an algorithm, Online Feasible Point method, that once followed by each agent, the coupled constraints of the game are never violated. The essence of the algorithm is that at the beginning of each round each agent $i$ transmits a desired set $S_t^{(i)}$. Then each agent $i$ selects an action $x_i^t \in S_t^{(i)}$ that depends also on the sets $S_t^{(j)}$ that the agent has received.
2. The authors identify a subclass of GNE i.e. strongly benign GNEP at which the proposed Online Feasible Point method converges to.
3. The authors establish no-regret guarantees for each individual agent that follows the Online Feasible Point method.

*Strengths*
The problem of how agents can collectively converge to a GNE is an interesting and important topic. Consequently, I believe that the results of this paper are valuable. The Online Feasible Point method guarantees feasibility at each step, which is a notable contribution compared to previous works. Additionally, the notion of strongly benign GNE is well-justified and is connected to standard assumptions in the learning-in-games literature, such as strong monotonicity.

*Weaknesses*
I believe the presentation of the paper could be significantly improved. Overall, I found the paper difficult to follow. Additionally, the authors should compare their convergence results with previous convergence results for GNE, such as those by Yi et al. and Jordan et al. It was unclear to me how the notion of strongly benign GNE as well as the provided convergence rates compare with the previous results. Furthermore, the no-regret guarantees seem to hold only in cases where a strongly benign GNEP exists. However, in my opinion, in order for an algorithm to be considered compatible with the strategic behavior, no-regret guarantees should hold regardless the game structure.

*Overall Evaluation*
I overall believe that this is an interesting paper and should be considered for acceptance.

**Paper Award:**

No

---

> ### Author Response · Authors · 2024-11-25
>
> We thank the reviewer for his insightful remarks and helpful feedback. Below, we address all your points in detail:
>
> Lacking clarity:
>
> We will do our best to improve the clarity of the paper. If the reviewer could point us to a concrete point in the paper that is unclear, we are happy to provide clarifications and integrate this in the paper.
>
> Comparison to previous results:
>
> We are not aware of any related work for which comparing convergence
> rates would make a fair comparison: we are the first to insist on strict
> feasibility per step, so by this metric no prior work succeeds. But to
> achieve strict feasibility we make quite different assumptions compared
> to prior work, so there is no natural comparison point in the literature
> that already provides convergence rates under our assumptions without
> requiring strict feasibility.
> See Section 2.2 for a comparison of our assumptions to
> more standard assumptions in the literature.
>
>
>
> Compatibility with Strategic Behavior without Strongly Benign GNEP:
>
> We agree with the reviewer that it is a limitation of our results that
> we only provide regret guarantees under certain conditions, so we cannot
> guarantee that our algorithm is fully robust against all kinds of
> strategic behavior. Our motivation to still discuss the relation to
> regret minimization was that: a) it does imply a limited form of
> robustness to strategic behavior by showing that no player would benefit
> in retrospect from deviating to any single fixed action; and b) it
> allows us to make a formal connection to the literature on online
> learning with time-varying constraint sets. Would it satisfy the
> reviewer if add a discussion of the desirability of unconditional regret
> guarantees to the future work?

---

### Author Rebuttal · Authors · 2024-11-25

We thank all reviewers for their helpful feedback, comments, and questions. We address all points separately in the comments section below.

---

### Meta-Review · Area_Chair_pekU · 2024-12-14

**Recommendation:** Accept
**Confidence:** 4

**Metareview:**

I agree with all the reviewers that this paper provides a nice contribution and it starts a a new direction by exploring GNEP in the online setting and for this reason I recommend acceptance.

Nevertheless, I also agree with Reviewer KU8B that the discussion about the information that needs to be exchanged is not sufficient from a computational perspective. The authors replied by arguing that there is no computational issue if $S^{(i)}_t$ can be expressed as a convex hull of a set of points, but this is not a very natural assumption and it does not capture the type of constraints that usually arise in market equilibrium problems that are formulated as GNEP. Usually in these settings the constraints $S^{(i)}_t$ are expressed in the form $g_i(x) \le 0$ for some convex function $g_i$. In this case, it is unclear what it means that one player should send $g_i$ to the others and whether this can be done in a computationally efficient way. For example, if we just need to be able to project to $S^{(i)}_t$ then a gradient oracle is sufficient to run a projection algorithm via the ellipsoid method, would such a gradient oracle be enough information for one player to send to the other?

As I said, despite the aforementioned weakness, I still believe that the paper provides an interesting first step in this important problem and I recommend acceptance. But, I expect from the authors to add more details to the final version about the computational issues that were raised by the reviewers and to spend enough time to polish their paper and address all the presentation issues that were raised by the reviewers.

**Paper Award:**

No